# Fam102a translocates Runx2 and Rbpjl to facilitate Osterix expression and bone formation

Yu Yamashita [1,2,5], Mikihito Hayashi [1,5] ✉, Anhao Liu [1,5], Fumiyuki Sasaki [1], Yosuke Tsuchiya[1], Hiroshi Takayanagi [3], Mitsuru Saito[2] & Tomoki Nakashima [4] ✉

Bone remodeling maintains the robustness of the bone tissue by balancing bone resorption by osteoclasts and bone formation by osteoblasts. Although these cells together play a crucial role in bone remodeling, only a few reports are available on the common factors involved in the differentiation of the two types of cells. Here, we show family with sequence similarity 102 member A (Fam102a) as a bone-remodeling factor that positively regulates both osteoclast and osteoblast differentiation. Fam102a regulates osteoblast differentiation by controlling recombination signal binding protein for immunoglobulin κ J region-like (Rbpjl). The Fam102a-Rbpjl axis promotes the nuclear translocation of transcription factors and enhances the expression of Osterix, a transcription factor essential for osteoblast differentiation. The deletion of *Fam102a* or a functional mutation in *Rbpjl* leads to osteopenia accompanied by reduced osteoblastic bone formation. Thus, the Fam102a-Rbpjl axis plays an important role in osteoblasts and this finding provides insights into bone remodeling.

Osteoporosis is an escalating medical and economic problem caused by an imbalance in bone homeostasis, which increases the risk of fragility fractures that, in turn, deleteriously impact ambulatory activities and, hence, cause quality of life issues[1,2]. Osteoporotic fractures typically occur in areas weakened by bone loss, such as the proximal femur and vertebrae of elderly patients. Even when treated surgically, fragility fractures are commonly associated with a decline in activities of daily living and an ultimate increase in mortality[2,3]. Osteoporotic fractures of the proximal femur and lumbar spine carry a 12-month excess mortality rate of up to 20% owing to an enhanced risk of complications[4]. Therefore, research on the mechanisms of bone metabolism is crucially required.

Bone is constantly renewed through a homeostatic process that balances resorption and formation. This "bone remodeling" process maintains the strength of bone tissue and tightly regulates calcium and phosphate metabolism[2,5]. The cells that constitute the bone, namely, osteoclasts, osteoblasts, and osteocytes, are mutually regulated by intercellular communication and disruption of this balance leads to various bone diseases, including osteoporosis[6–8]. Therefore, elucidating the regulatory mechanisms involved in bone remodeling is important for understanding both bone physiology and pathophysiology. Although several reports had dealt with the distinct regulatory mechanisms of osteoclast and osteoblast differentiation[6–8], relatively few reports have focused on the common factors involved in the differentiation of both cell types, partly because osteoclasts are derived from hematopoietic stem cells, and osteoblasts are derived from skeletal stem cells. Certain factors and pathways, such as nuclear factor of activated T cells (NFAT) and Wnt signaling, have been shown to

[1]Department of Cell Signaling, Graduate School of Medical and Dental Sciences, Institute of Science Tokyo, Tokyo, Japan. [2]Department of Orthopaedic Surgery, The Jikei University School of Medicine, Tokyo, Japan. [3]Department of Immunology, Graduate School of Medicine and Faculty of Medicine, The University of Tokyo, Tokyo, Japan. [4]Faculty of Dentistry, Institute of Science Tokyo, Tokyo, Japan. [5]These authors contributed equally: Yu Yamashita, Mikihito Hayashi, Anhao Liu. ✉e-mail: hayashi.csi@tmd.ac.jp; naka.csi@tmd.ac.jp

regulate both osteoclast and osteoblast differentiation. Wnt signaling is a potent activator of osteoblast differentiation and enhances osteoclast differentiation[7,9]. The transcription factor NFAT is a key regulator of osteoclast differentiation[10]; it also regulates osteoblast differentiation in cooperation with Osterix (Osx, encoded by *Sp7*)[11]. The identification of a novel bone-remodeling factor that regulates both osteoclast and osteoblast differentiation advances our understanding of the mechanisms of bone metabolism and supports the development of innovative osteoporosis therapies.

Therefore, we screened for a molecule that regulates both osteoclast and osteoblast differentiation as a crucially important bone remodeling factor and identified the family with sequence similarity 102 member A (*Fam102a*) gene by comprehensive gene expression analysis. When the breast cancer cell line MCF-7 was stimulated with estrogen, the expression of *Fam102a*, also known as early estrogen-induced gene 1 (*Eeig1*), was upregulated during the early phase[12]. In this study, we found that Fam102a dominantly enhances osteoblast differentiation by regulating the expression of Osx as a result of controlling the nuclear translocation of runt-related transcription factor 2 (Runx2). Runx2 and Osx are master regulators of osteoblast differentiation[7]. Fam102a plays an important role in the maintenance of bone mass by regulating bone formation in vivo.

We identified the transcription factor "recombination signal binding protein for immunoglobulin κ J region-like" (Rbpjl), which is a member of the Rbpj gene family, as a regulator of osteoblast differentiation by comprehensive gene expression analysis of *Fam102a*–deficient osteoblasts. Rbpjl is highly expressed in pancreatic adenocytes which are promoted by self-amplification via the Notch signaling pathway[13–15]. However, it remains unclear whether Rbpjl contributes to bone homeostasis by regulating osteoblast differentiation and whether it has a self-amplifying capacity in osteoblasts. In this study, we demonstrated that Rbpjl self-amplifies in osteoblasts and maintains bone mass by regulating the expression of Osx together with Runx2, independent of Notch signaling. In addition, auto-amplification of Rbpjl is regulated by Fam102a through nuclear translocation.

## Results

### Fam102a-mediated regulation of bone remodeling

We performed a comprehensive gene expression analysis of cells deficient in each key transcription factor to identify the common factors that regulate both osteoclast and osteoblast differentiation. We analyzed cells derived from mice deficient in nuclear factor of activated T cells 1 (*Nfatc1*), a key transcription factor in osteoclast differentiation, in osteoclasts (*Ctsk*^Cre/+ *Nfatc1*^flox/flox mice)[16–18] to narrow down the list of genes that might positively regulate osteoclasts. These mice exhibited a complete osteopetrotic phenotype at 3 weeks of age (Supplementary Fig. 1a, b) and stimulation of splenocytes derived from these mice with receptor activator of nuclear factor-κB ligand (RANKL) and macrophage colony-stimulating factor (M-CSF) did not induce tartrate-resistance acid phosphatase (TRAP)−positive multinucleated cell formation (Supplementary Fig. 1c). In cells derived from *Ctsk*^Cre/+ *Nfatc1*^flox/flox mice, the expression of various genes related to osteoclast differentiation decreased (Supplementary Fig. 2a). We identified 18 genes that were highly expressed in mature osteoclasts, upregulated upon differentiation, and downregulated in cells from *Ctsk*^Cre/+ *Nfatc1*^flox/flox mice compared to that in *Ctsk*^Cre/+ *Nfatc1*^flox/+ cells. We examined cells derived from *Runx2*–deficient mice to identify the genes that potentially regulate osteoblast differentiation. Our analysis revealed the downregulation of genes associated with osteoblast differentiation in *Runx2*–deficient osteoblasts (Supplementary Fig. 2b). We identified 52 genes that were highly expressed in mature osteoblasts, were upregulated during differentiation, and were downregulated in *Runx2*–deficient osteoblasts. Of these, *Fam102a* was identified as the only gene common to both types of bone cells

(Fig. 1a). Fam102a was expressed in a wide range of cells, and among the various cell types analyzed, we found that it was expressed in osteoblasts and at high levels in osteoclasts (Supplementary Fig. 3).

We analyzed the expression and function of Fam102a in vitro to determine whether *Fam102a* regulates the differentiation of osteoclasts and osteoblasts. *Fam102a* expression was upregulated in both osteoclasts and osteoblasts during differentiation (Fig. 1b, c). As Fam102a expression was shown to be under the control of NFATc1 signaling in osteoclasts (Fig. 1a), we analyzed the involvement of NFATc1 in *Fam102a* expression in osteoblasts. Treatment of the osteoblast cell line MC3T3-E1 with the calcineurin inhibitor INCA-6, which inhibits NFAT activation, did not affect *Fam102a* expression (Supplementary Fig. 4a). As *Fam102a* was identified as a gene with expression upregulated early after estrogen treatment in MCF-7[12], we analyzed the effect of estrogen treatment on *Fam102a* expression in both osteoclasts and osteoblasts. However, estrogen treatment had no effect on *Fam102a* expression in preosteoclasts or preosteoblasts (Supplementary Fig. 4b, c). These results indicate that estrogen signaling is not involved in *Fam102a* expression in either cell type.

As Fam102a has been reported to play a role in osteoclasts[19,20], we examined its role in osteoblasts. The lentiviral knockdown of Fam102a inhibited osteoblast differentiation (Fig. 1d). Analysis of osteoblastic gene expression in *Fam102a* knockdown cells revealed that the expression of *Runx2*, a transcription factor essential for osteoblast differentiation, was unchanged, whereas the expression of *Sp7*, another transcription factor essential for osteoblast differentiation, was decreased (Fig. 1e). Conversely, overexpression of Fam102a in MC3T3-E1 cells promoted osteoblast differentiation and increased *Sp7* expression, but not *Runx2* expression (Supplementary Fig. 5a–c). These results suggest that Fam102a regulates osteoblast differentiation downstream of Runx2. In fact, overexpression of Runx2 significantly induced the expression of *Fam102a* (Fig. 1f). To examine whether Runx2 directly regulates *Fam102a* expression, we confirmed the Runx2 binding site in the *Fam102a* promoter region based on the available ChIP-Seq database (ChIP-Atlas[21]) using a luciferase assay of gene promoter activity. Analysis of the Runx2 binding site in the promoter region of *Fam102a* showed that no Fam102a reporter signal was induced by Runx2 expression (Supplementary Fig. 5d, e). In contrast, examination of the enhancer region of *Fam102a* to which Runx2 binds (Supplementary Fig. 5d), showed that Runx2 had enhancer activity for *Fam102a* (Fig. 1g). Furthermore, deletion of Runx2 binding sites within the enhancer region abolished Runx2-mediated transactivation of *Fam102a* (Supplementary Fig. 5d, f). These results suggest that the recruitment of Runx2 to the enhancer region of *Fam102a* positively regulates the expression of *Fam102a*.

Several studies have reported that Fam102a in osteoclasts is predominantly localized to the cytoplasm[19] or nucleus[20]. To clarify the localization of Fam102a in osteoblasts, we examined the localization of Fam102a protein fused with turbo green fluorescent protein (tGFP) expressed in MC3T3-E1 cells and found that Fam102a-tGFP was localized in the cytoplasm and nucleus (Fig. 1h). These results indicated that Fam102a is induced by Runx2 activation and is involved in osteoblast differentiation.

### Mice with conditional deletion of *Fam102a* in osteoclasts

We analyzed cell-specific *Fam102a*–deficient mice to elucidate the in vivo role of Fam102a in bone cells. Osteoclast-specific *Fam102a*–deficient mice were generated by crossing *Ctsk*^Cre/+ mice with *Fam102a*-floxed mice (Supplementary Fig. 6a–c). The mice survived and developed normally. Micro-computed tomography (μCT) along with bone morphometric analyses of the distal femur and the proximal tibia revealed that male *Ctsk*^Cre/+ *Fam102a*^flox/Δ mice had significantly increased trabecular bone volume compared to *Fam102a*^flox/Δ mice due to lower osteoclastic bone resorption and comparable levels of osteoblastic bone formation at 12 weeks of age (Fig. 2a–d). To assess

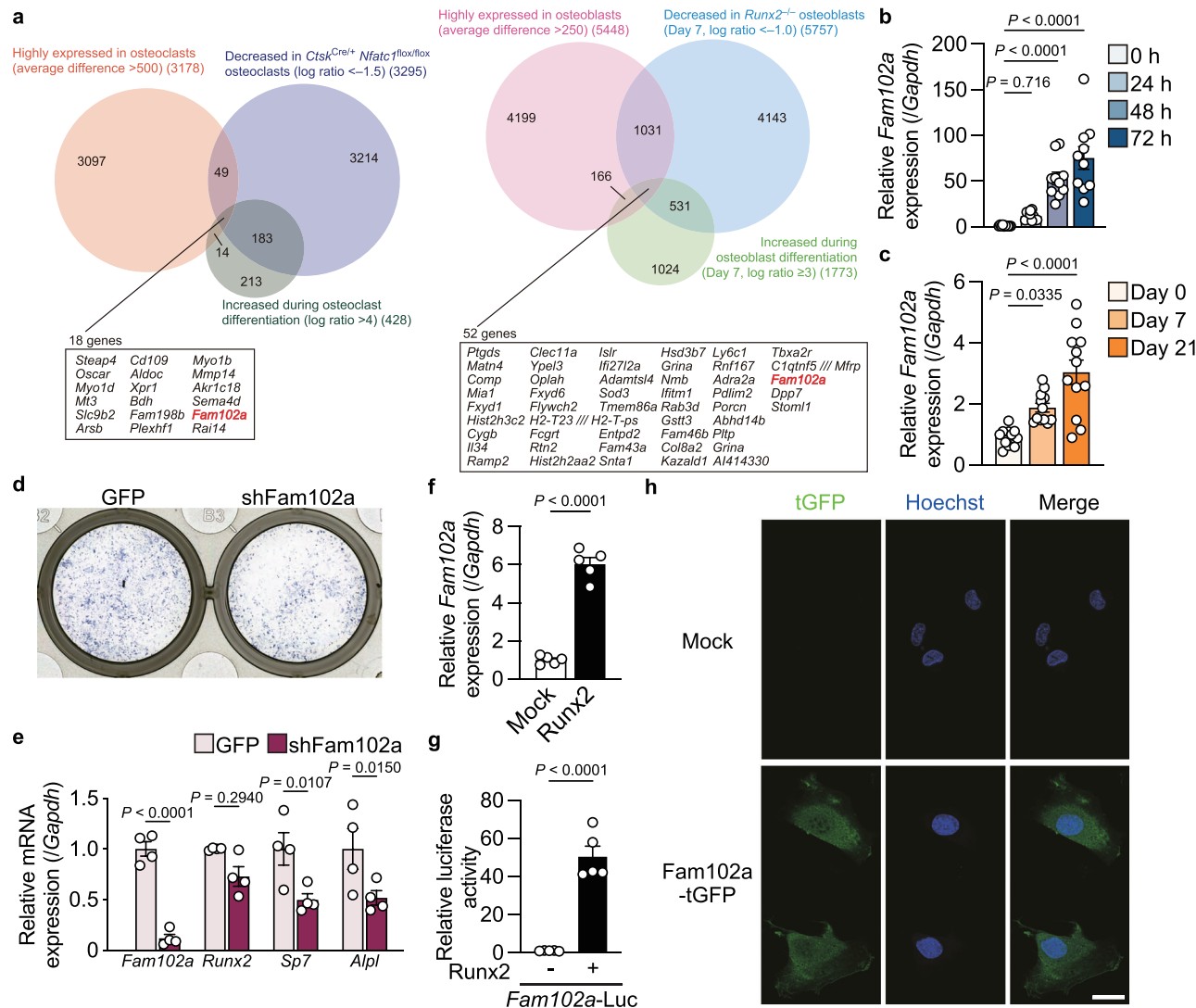

**Fig. 1 | Identification of Fam102a as a regulator of bone remodeling.**
**a** Identification of Fam102a as a Runx2- and NFATc1-induced gene. **b, c** *Fam102a* mRNA expression during osteoclast (**b**, *n* = 10) and osteoblast (**c**, Day 0: *n* = 11, Day 7 and 21: *n* = 12) differentiation. **d** ALP staining of turboGFP (tGFP) and shFam102a expressing calvarial cells cultured in osteogenic medium for 7 days. **e** The effect of lentiviral knockdown of Fam102a on osteoblastic gene expression in calvarial cells cultured in osteogenic medium for 7 days (*n* = 4; independent experimental replicates). **f** The effect of retroviral overexpression of Runx2 on *Fam102a* mRNA expression in MC3T3-E1 cells cultured in osteogenic medium for 7 days (*n* = 5;

independent experimental replicates). **g** The effect of Runx2 expression on the reporter activity of *Fam102a*-Luc (region 2) in MC3T3-E1 cells (*n* = 5; independent experimental replicates). **h** Fam102a localization in MC3T3-E1 cells lentivirally expressing Fam102a-tGFP (green). Nuclei were stained with Hoechst 33342 (blue) (scale bar, 20 μm). Three independent experiments were performed. Data are shown as the mean ± SEM. Statistical analyses were performed using one-way ANOVA and Tukey's *post hoc* test (**b** and **c**) or unpaired two-sided Student's *t* test (**e**–**g**). Source data are provided as a Source Data file.

the impact of *Fam102a* deficiency in *Ctsk*⁺ cells on cortical bone, we examined the cortical bone parameters of the distal femur using μCT. At 12 weeks of age, the cortical bone thickness, volume, and periosteal perimeter of male *Ctsk*^Cre/+ *Fam102a*^flox/Δ mice were comparable to those of the control mice (Supplementary Fig. 7). Osteoclast differentiation from bone marrow-derived macrophages (BMMs) decreased in *Ctsk*^Cre/+ *Fam102a*^flox/Δ mice (Fig. 2e). These results suggest that Fam102a is involved in bone remodeling by promoting osteoclast differentiation in vivo, which is consistent with a previous report[19].

**Mice with conditional deletion of *Fam102a* mice in osteoblasts**
To investigate the role of Fam102a in osteoblasts, we analyzed mice in which *Fam102a* was specifically deleted in *Sp7-Cre*⁺ cells, including osteoblasts. Analysis of the distal femur and proximal tibia of male *Sp7-Cre*⁺ *Fam102a*^flox/Δ mice revealed a significant decrease in trabecular

bone with suppressed osteoblastic bone formation and comparable levels of osteoclastic bone resorption at 12 weeks of age (Fig. 3a–d). μCT analysis revealed that cortical bone thickness and volume were significantly reduced in male *Sp7-Cre*⁺ *Fam102a*^flox/Δ mice, whereas periosteal perimeters were comparable to those in *Sp7-Cre*⁺ *Fam102a*^flox/+ mice (Supplementary Fig. 8). These results indicated that Fam102a is involved in bone remodeling by promoting osteoblast differentiation in vivo.

**Osteoporotic phenotype of global *Fam102a*–deficient mice**
Although we have demonstrated that Fam102a positively regulates both osteoclast and osteoblast differentiation by analyzing mouse strains with respective cell-specific deficiencies, the comprehensive role of Fam102a in vivo remains unclear. We generated mice lacking *Fam102a* throughout the body (*Fam102a*^Δ/Δ) by crossing *Actb-Cre* mice

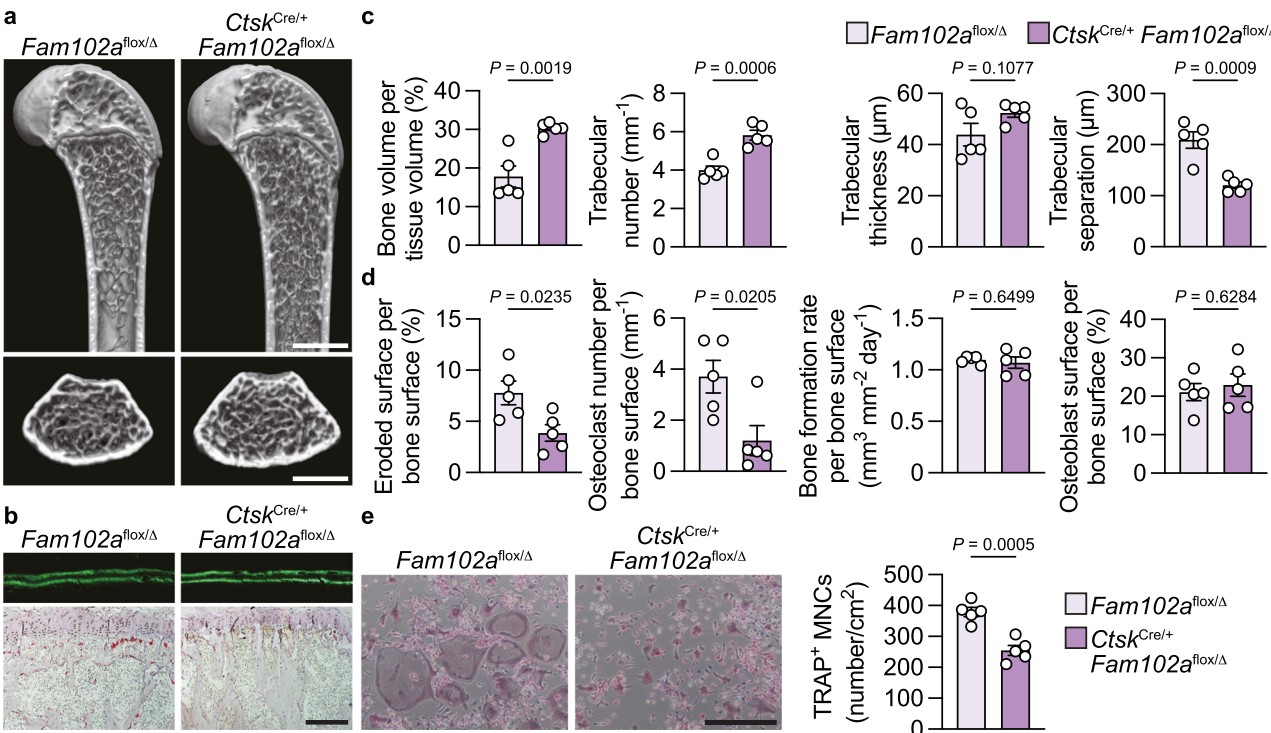

**Fig. 2 | Mice wih conditional deletion of *Fam102a* in osteoclasts exhibit an osteopetrotic phenotype. a** Representative micro-computed tomography (μCT) images of the distal femur in 12-week-old male *Fam102a*^flox/Δ and *Ctsk*^Cre/+ *Fam102a*^flox/Δ mice (scale bar, 1 mm). **b** New bone formation was determined by calcein double labeling (upper panel). TRAP staining of the proximal tibiae of 12-week-old male *Fam102a*^flox/Δ and *Ctsk*^Cre/+ *Fam102a*^flox/Δ mice (lower panel) (scale bar, 0.2 mm). **c** μCT analysis of the distal femur in 12-week-old male *Fam102a*^flox/Δ and *Ctsk*^Cre/+ *Fam102a*^flox/Δ mice (*n* = 5). **d** Bone morphometric analysis in the proximal tibiae in 12-week-old male *Fam102a*^flox/Δ and *Ctsk*^Cre/+ *Fam102a*^flox/Δ mice (*n* = 5). **e** In vitro osteoclast differentiation from *Fam102a*^flox/Δ and *Ctsk*^Cre/+ *Fam102a*^flox/Δ BMMs analyzed by TRAP staining (scale bar, 0.4 mm) (*n* = 5). Data are shown as the mean ± SEM. Statistical analyses were performed using unpaired two-sided Student's *t* test. Source data are provided as a Source Data file.

with *Fam102a*-floxed mice to determine the gross role of Fam102a in the bone in vivo. The body lengths and weights of 12-week-old *Fam102a*^Δ/Δ mice were similar to those of control mice. Bone analysis revealed a significant reduction in trabecular and cortical bone in the distal femur and proximal tibia of male *Fam102a*^Δ/Δ mice, along with suppressed osteoblastic bone formation and osteoclastic bone resorption (Fig. 4a–e). Similarly, the trabecular bone in the distal femur was significantly reduced in female *Fam102a*^Δ/Δ mice at 12 weeks of age (Supplementary Fig. 9a). We also examined the phenotypes of organs other than bones in *Fam102a*^Δ/Δ mice. Despite a significant decrease in pancreatic weight and an increase in lung weight among the major organs, no apparent abnormalities were observed in fasting blood glucose levels and histological analysis of the pancreas and lungs in 12-week-old male *Fam102a*^Δ/Δ mice (Supplementary Fig. 9b, c, and Supplementary Table 1). Therefore, we conclude that these changes in organ weight did not affect bone metabolism. Given that Fam102a is predominantly involved in bone formation and that *Fam102a*^Δ/Δ mice show reduced osteoblast and osteoclast differentiation, *Fam102a*^Δ/Δ mice exhibited an osteoporosis-like phenotype with low bone turnover at 12 weeks of age. Osteoblast differentiation in *Fam102a*–deficient calvarial cells was significantly reduced compared to that in control cells (Fig. 4f, g). Gene expression analysis showed that *Runx2* expression remained unchanged, whereas *Sp7* expression was significantly decreased (Fig. 4h), similar to the results obtained using knockdown cells. These results suggested that Fam102a controls *Sp7* expression rather than *Runx2* expression in osteoblasts. As previously reported, osteoclast differentiation of BMMs derived from *Fam102a*^Δ/Δ mice was decreased (Supplementary Fig. 10a) and analysis of osteoclastic gene expression showed a significant reduction in *Nfatc1* expression (Supplementary Fig. 10b).

## Fam102a enhances Runx2 activity by regulating nuclear translocation of Kpna2

To explore the mechanism by which Fam102a regulates *Sp7* expression, we performed total RNA-seq analysis of osteoblasts differentiated from *Fam102a*–deficient calvarial cells (Fig. 5a). We confirmed that genes associated with osteoblast differentiation were downregulated in osteoblasts lacking Fam102a (Supplementary Fig. 11a). Gene ontology (GO) analysis of the differentially expressed genes using DAVID[22] revealed a significant decrease in terms related to osteogenesis and cell proliferation (Fig. 5b). GO (i.e. biological process) pathway analysis of the terms for cell proliferation in QuickGO[23] showed that the most downstream term was "positive regulation of cell proliferation" (Supplementary Fig. 11b). Protein-protein interaction (PPI) network analysis of a group of genes belonging to this term using STRING[24] showed that fibroblast growth factor 2 (FGF2) was strongly correlated (Supplementary Fig. 11c). In fact, FGF2 treatment of osteoblastic cells from *Fam102a*^Δ/Δ mice resulted in decreased phosphorylation of ERK and p38 compared to that in cells from wild-type mice (Supplementary Fig. 11d). In contrast, osteoblastic cells from *Fam102a*^Δ/Δ mice showed no changes in cell proliferation without FGF2 stimulation (Supplementary Fig. 11e). Collectively, despite reduced FGF2 signaling activity, cell proliferation was not affected in *Fam102a*^Δ/Δ osteoblasts, suggesting that FGF2 signaling activity has minor effect on osteoblast differentiation in cells derived from *Fam102a*^Δ/Δ mice.

The "positive regulation of cell proliferation" included 13 genes (Fig. 5a), of which the expression of *Fgfr2*, *Fgfr3*, and *Htra1* is known to be regulated by Runx2[25,26], an essential transcription factor involved in both osteogenesis and cell proliferation in osteoblasts[27]. Based on these findings, we hypothesized that Runx2 function is attenuated in *Fam102a*–deficient osteoblasts.

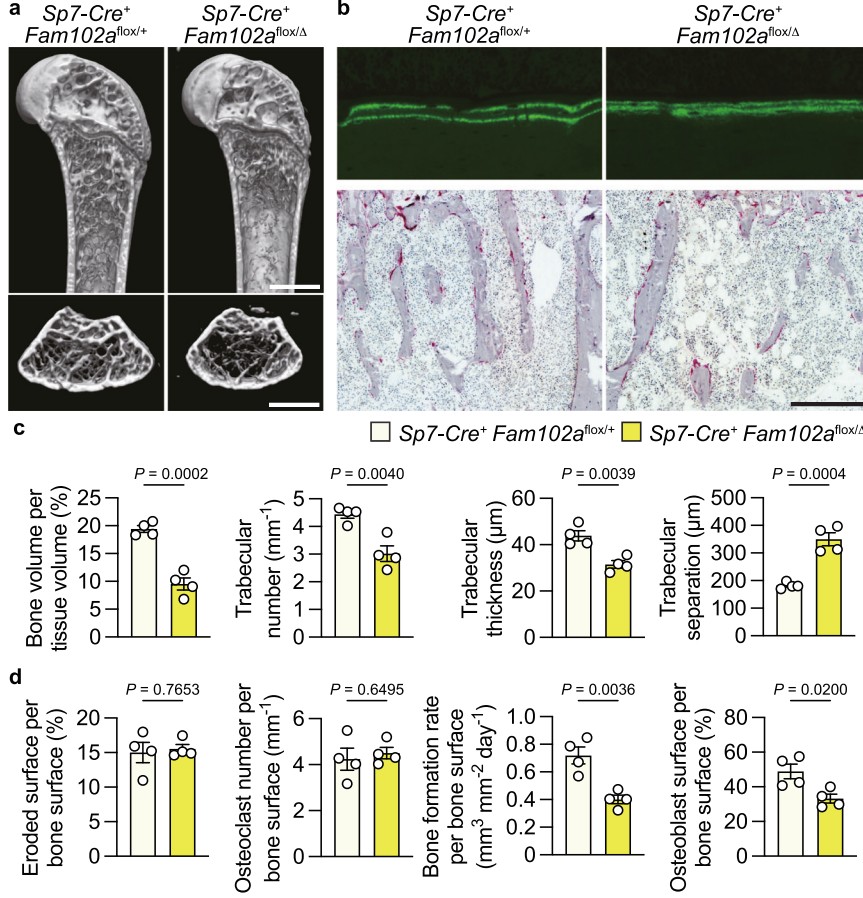

**Fig. 3 | Mice with conditional deletion of *Fam102a* in osteoblast lineage cells exhibit an osteoporotic phenotype. a** Representative µCT images of the distal femur in 12-week-old male *Sp7-Cre⁺ Fam102a^flox/+^* and *Sp7-Cre⁺ Fam102a^flox/Δ^* mice (scale bar, 1 mm). **b** New bone formation was determined by calcein double labeling (upper panel). TRAP staining of the proximal tibiae of 12-week-old male *Sp7-Cre⁺ Fam102a^flox/+^* and *Sp7-Cre⁺ Fam102a^flox/Δ^* mice (lower panel) (scale bar, 0.2 mm).

**c** µCT analysis of the distal femur in 12-week-old male *Sp7-Cre⁺ Fam102a^flox/+^* and *Sp7-Cre⁺ Fam102a^flox/Δ^* mice ($n = 5$). **d** Bone morphometric analysis in the proximal tibiae in 12-week-old male *Sp7-Cre⁺ Fam102a^flox/+^* and *Sp7-Cre⁺ Fam102a^flox/Δ^* mice ($n = 5$). Data are shown as the mean ± SEM. Statistical analyses were performed using unpaired two-sided Student's *t* test. Source data are provided as a Source Data file.

In a reporter assay of the *Sp7* promoter region (region 1; Supplementary Fig. 12a), which contains a Runx2 binding site that has been previously reported to have promoter activity[28,29], no change in *Sp7* promoter activity was observed in the presence of Runx2 (Supplementary Fig. 12b). Conversely, *Sp7* promoter activity was significantly increased in the presence of Runx2 in reporter assays of the 12 kb upstream region of *Sp7* (region 2), which includes multiple Runx2 binding regions (Fig. 5c)[30]. Furthermore, Fam102a overexpression increased *Sp7* promoter activation by Runx2 (Fig. 5c). Thus, Fam102a altered *Sp7* expression by regulating Runx2 activity. Notably, lentiviral overexpression of Runx2 in *Fam102a*−deficient osteoblasts rescued the impaired osteoblast differentiation (Supplementary Fig. 12c, d).

As Runx2 protein expression was normal in osteoblastic cells from *Fam102a^Δ/Δ^* mice (Fig. 5d), we investigated the possibility of altered subcellular localization of Runx2 as a mechanism by which Fam102a regulates Runx2 activity. We found that the nuclear localization of Runx2 decreased in osteoblastic cells from *Fam102a^Δ/Δ^* mice (Fig. 5e). Similar results were obtained by immunofluorescence staining for Runx2 (Fig. 5f). As these results suggested that Fam102a may be involved in the nuclear translocation of Runx2, we performed a co-immunoprecipitation (Co-IP) assay to examine the interaction of Fam102a with importins that regulate nuclear trafficking. We observed an interaction between Fam102a and the importin subunit karyopherin subunit alpha 2 (Kpna2), which is highly expressed in osteoblastic cells

(Fig. 5g, h). Furthermore, our analysis showed that Fam102a and Runx2 did not interact directly, but rather through Kpna2 (Fig. 5i). To identify the binding regions, we generated mutants of both Fam102a and Kpna2, which revealed that Fam102a binds to the armadillo repeat domain of Kpna2 and that both the N- and C-terminal regions of Fam102a are necessary for this interaction (Supplementary Fig. 13a, b). In addition, we evaluated the binding ability of Fam102a to Runx2 binding partners, including yes-associated protein 1 (Yap1), transcriptional coactivator with PDZ-binding motif (Taz), hes family bHLH transcription factor 1 (Hes1), and signal transducer and activator of transcription 1 (Stat1), which have been shown to directly interact with Runx2 and regulate its transcriptional activity[31–34]. Co-IP analysis did not show any interaction between these molecules and Fam102a (Supplementary Fig. 13c). These results indicate that Fam102a regulates Runx2 activity by binding to Kpna2, thereby controlling Runx2 nuclear translocation.

### Identification of Rbpjl as an osteoblast regulatory factor

Despite a partial (approximately 75%) reduction in the nuclear localization of Runx2 (Fig. 5e), *Fam102a^Δ/Δ^* mice showed a significant decrease in bone mass (Fig. 4a). Therefore, we hypothesized that additional mechanisms beyond the reduction in the nuclear localization of Runx2 may contribute to the impaired osteoblast differentiation induced by Fam102a deficiency. As Fam102a potentially

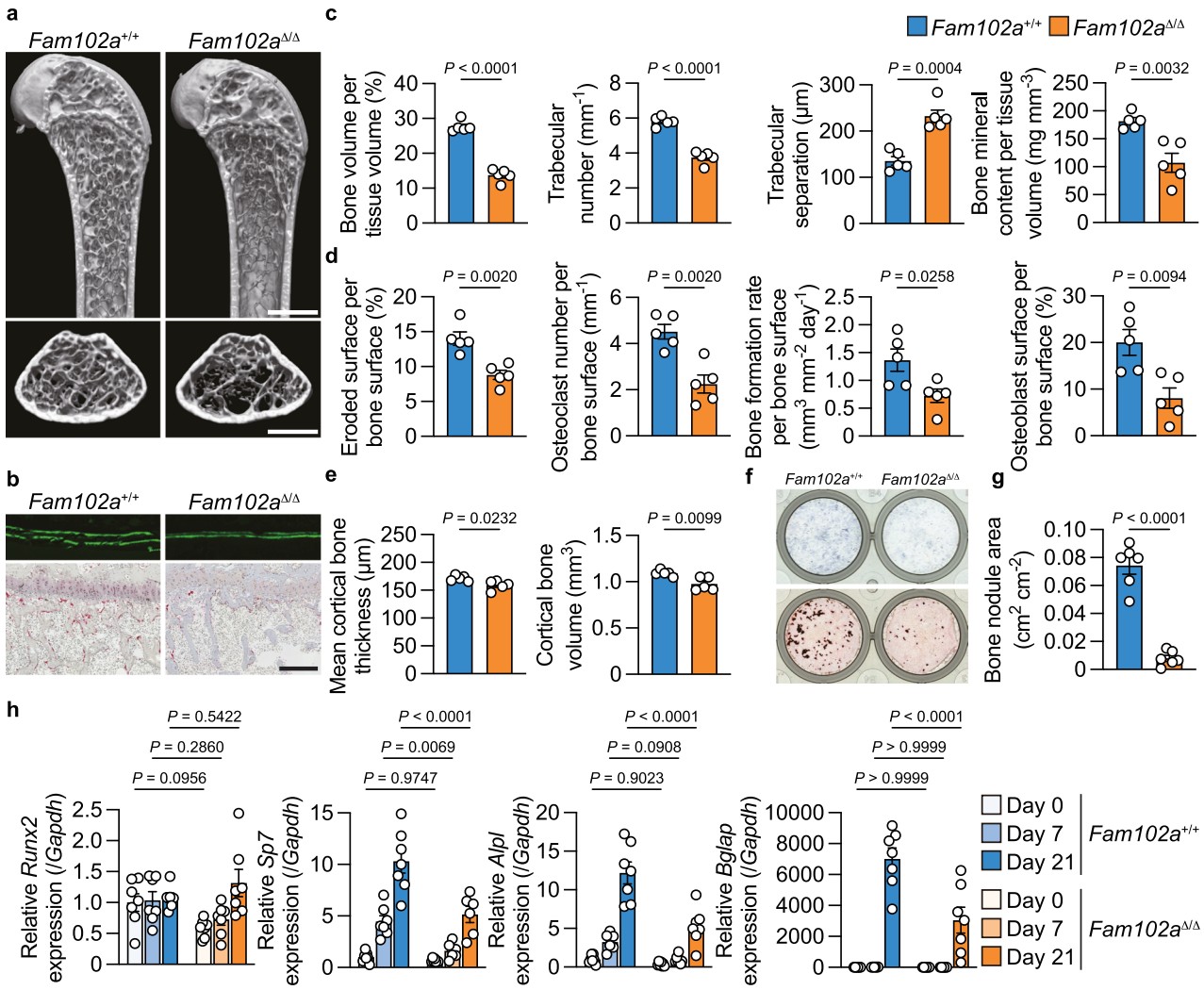

**Fig. 4 | Mice lacking *Fam102a* throughout the body exhibit an osteoporotic phenotype. a** Representative μCT images of the distal femur in 12-week-old male *Fam102a*^Δ/Δ mice and their littermate controls (scale bar, 1 mm). **b** New bone formation was determined by calcein double labeling (upper panel). TRAP staining of the proximal tibiae of 12-week-old male *Fam102a*^Δ/Δ mice and their littermate controls (lower panel) (scale bar, 0.2 mm). **c** μCT analysis of the distal femur in 12-week-old male *Fam102a*^Δ/Δ mice and their littermate controls (*n* = 5). **d** Bone morphometric analysis in the proximal tibiae of 12-week-old male *Fam102a*^Δ/Δ mice and their littermate controls (*n* = 5). **e** Cortical bone analysis as measured by μCT in 12-week-old male *Fam102a*^Δ/Δ mice and their littermate controls (*n* = 5). **f** In vitro

osteoblast differentiation of *Fam102a*^+/+ and *Fam102a*^Δ/Δ calvarial cells shown by ALP staining (at 7 days of culture, upper panel) and Alizarin Red S staining (at 21 days of culture, lower panel). **g** Bone nodule formation in *Fam102a*^+/+ and *Fam102a*^Δ/Δ calvarial cells cultured in osteogenic medium for 21 days (*n* = 6; independent experimental replicates). **h** mRNA expression of osteoblastic genes during differentiation in *Fam102a*^+/+ and *Fam102a*^Δ/Δ calvarial cells (*n* = 7; independent experimental replicates). Data are shown as the mean ± SEM. Statistical analyses were performed using unpaired two-sided Student's *t* test (**c**–**e** and **g**) or two-way ANOVA and Sidak's *post hoc* test (**h**). Source data are provided as a Source Data file.

participates in the nuclear translocation of transcription factors other than Runx2 by binding to Kpna2 in osteoblasts, we focused on transcription factors with altered expression in *Fam102a*–deficient osteoblastic cells. By comprehensively profiling the expression of transcription factors in *Fam102a*–deficient osteoblasts, we identified *Rbpjl* as the most downregulated transcription factor in the absence of Fam102a (Fig. 6a). *Rbpjl*, a member of the *Rbpj* gene family of downstream target genes of the Notch signaling pathway, is reported to be highly expressed and self-amplified in pancreatic adenocytes[13–15]. We focused on the possibility that Fam102a contributes to the self-amplification ability of Rbpjl and found not only reduced expression of Rbpjl mRNA and protein in osteoblastic cells from *Fam102a*^Δ/Δ mice (Fig. 6b and Supplementary Fig. 14a), but also an interaction between Rbpjl and Fam102a (Fig. 6c). In MC3T3-E1 cells stably expressing Rbpjl, knockdown of Fam102a expression reduced Rbpjl nuclear expression

(Fig. 6d). These results indicate that Fam102a knockdown suppresses the nuclear translocation and self-amplification of Rbpjl, thereby reducing *Rbpjl* expression. Rbpjl did not bind to importin α, suggesting that Fam102a is required for nuclear translocation of Rbpjl (Supplementary Fig. 14b). Changes in Notch family gene expression during osteoblast differentiation revealed that *Rbpjl* was significantly upregulated after the induction of osteoblastic differentiation (Supplementary Fig. 14c). Based on these results, we investigated the role of Rbpjl in osteoblasts, as Rbpjl functions as a transcription factor downstream of Fam102a and may be involved in osteoblast differentiation. The subcellular localization of Rbpjl in osteoblastic cells was examined and found to be similar to that of Rbpj, which is localized in the nucleus[35] (Fig. 6e, f). These results suggest that Rbpjl, similar to Rbpj, may regulate osteoblast differentiation. Furthermore, we found that the knockdown of Rbpjl in primary calvarial cells suppressed

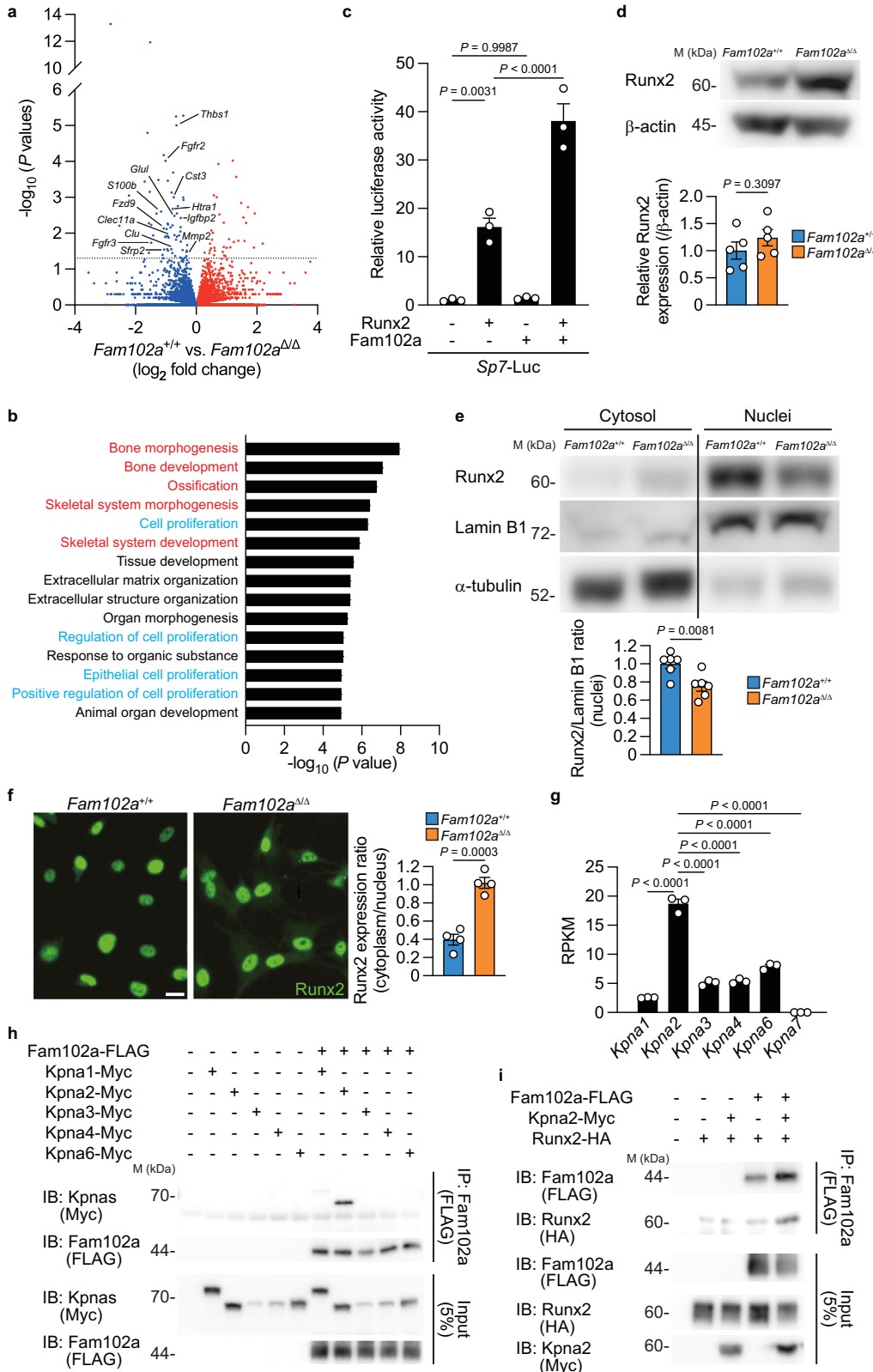

osteoblast differentiation (Fig. 6g). In these cells, as in *Fam102a*-deficient osteoblastic cells, *Runx2* expression was unchanged, whereas *Sp7* expression decreased (Fig. 6h). In contrast, Rbpjl overexpression in MC3T3-E1 cells did not affect osteoblast differentiation (Supplementary Fig. 15a–c). These results suggest that Rbpjl, which is relatively poorly expressed in osteoblasts, is not required at high levels.

To elucidate the role of Rbpjl in vivo, *Rbpjl*-mutated mice carrying an in-frame mutation with a 42 bp deletion including the DNA-binding domain[13,15] were generated using the *i*-GONAD method (Supplementary Fig. 16a)[36]. Western blotting of primary osteoblastic cells from these mice revealed a decrease in the size of Rbpjl protein (Supplementary Fig. 16b). Male Rbpjl-mutated mice had normal body size and, as previously reported[15], reduced pancreatic weight; but, no effect was

**Fig. 5 | Fam102a enhances Runx2 activity by regulating nuclear translocation with Kpna2. a** Scatter plot of the RNA-seq profiles of *Fam102a^{Δ/Δ}* osteoblastic cells versus *Fam102a^{+/+}* osteoblastic cells cultured in osteogenic medium for 7 days (*n* = 3). The dotted line indicates statistically significant values (*P* = 0.05). **b** Gene ontology (Biological process) enrichment analysis of genes differentially decreased in *Fam102a^{Δ/Δ}* osteoblastic cells relative to *Fam102a^{+/+}* osteoblastic cells cultured in osteogenic medium for 7 days (*n* = 3). Terms related to bone formation and cell proliferation are shown in red and cyan, respectively. **c** The effect of Runx2 and Fam102a expression on the reporter activity of *Sp7*-Luc (region 2) in MC3T3-E1 cells (*n* = 3; independent experimental replicates). **d** Upper panel, representative western blot images of Runx2 and β-actin in *Fam102a^{+/+}* and *Fam102a^{Δ/Δ}* calvarial cells cultured in osteogenic medium for 7 days. Lower panel, quantified western blot results (*n* = 5; independent experimental replicates). **e** Upper panel, representative western blot images of nuclear and cytoplasmic Runx2, Lamin B1 and α-tubulin in *Fam102a^{+/+}* and *Fam102a^{Δ/Δ}* calvarial cells cultured in osteogenic medium for

7 days. Lower panel, quantified western blot analysis of nuclear Runx2 (*n* = 6; independent experimental replicates). **f** Left panel, representative images of immunofluorescence staining of Runx2 in *Fam102a^{+/+}* and *Fam102a^{Δ/Δ}* calvarial cells cultured in osteogenic medium for 7 days (scale bar, 20 μm). Right panel, quantification of immunofluorescence staining of Runx2 expression (*n* = 4; independent experimental replicates). **g** The mRNA expression of *Kpna* family analyzed by RNA-seq in wild-type calvarial cells (*n* = 3). RPKM, Reads Per Kilobase of exon per Million mapped reads. **h** The analysis of the interaction between Fam102a and Kpna family members examined by co-immunoprecipitation (Co-IP) assay. Three independent experiments were performed. **i** The analysis of the interaction between Runx2, Fam102a and Kpna2 examined by Co-IP assay. Three independent experiments were performed. M, molecular mass. Data are shown as the mean ± SEM. Statistical analyses were performed using unpaired two-sided Student's *t* test (**a, d, e, f**), Fisher's exact test (**b**) or one-way ANOVA and Tukey's *post hoc* test (**c** and **g**). Source data are provided as a Source Data file.

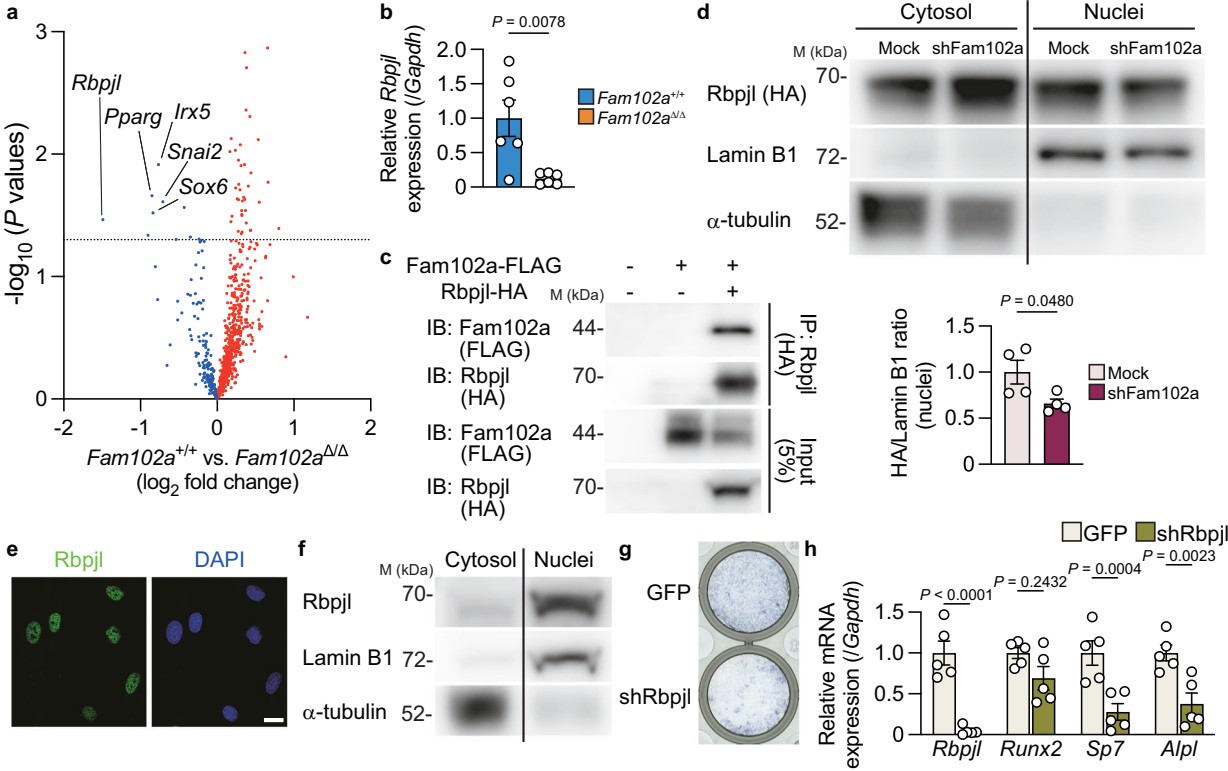

**Fig. 6 | Identification of Rbpjl as an osteoblast regulatory factor. a** Scatter plot of RNA-seq analysis of transcription factors in *Fam102a^{Δ/Δ}* osteoblastic cells versus *Fam102a^{+/+}* osteoblastic cells cultured in osteogenic medium for 7 days (*n* = 3). The dotted line indicates statistically significant values (*P* = 0.05). **b** *Rbpjl* mRNA expression in *Fam102a^{+/+}* and *Fam102a^{Δ/Δ}* calvarial cells cultured in osteogenic medium for 7 days (*n* = 6; independent experimental replicates). **c** The analysis of the association between Fam102a and Rbpjl examined by Co-IP assay. Three independent experiments were performed. **d** Upper panel, representative western blot images of nuclear and cytoplasmic Rbpjl, Lamin B1 and α-tubulin in MC3T3-E1 cells stably expressing HA-tagged Rbpjl cultured in osteogenic medium for 7 days after Fam102a knockdown. Lower panel, quantified western blot analysis of nuclear Rbpjl-HA (*n* = 4; independent experimental replicates). **e** Representative images of

immunofluorescence staining of Rbpjl in wild-type calvarial cells (scale bar, 20 μm). Three independent experiments were performed. **f** Representative western blot images of nuclear and cytoplasmic Rbpjl, Lamin B1 and α-tubulin in wild-type calvarial cells cultured in osteogenic medium for 7 days. Three independent experiments were performed. **g** ALP staining of calvarial cells expressing GFP and shRNA against Rbpjl cultured in osteogenic medium for 7 days. **h** The effect of lentiviral knockdown of Rbpjl on osteoblastic gene expression in calvarial cells cultured in osteogenic medium for 7 days (*n* = 5; independent experimental replicates). M, molecular mass. Data are shown as the mean ± SEM. Statistical analyses were performed using unpaired two-sided Student's *t* test (**a, b, d, h**). Source data are provided as a Source Data file.

observed on glucose metabolism at 12 weeks of age (Supplementary Fig. 16c–e). Accordingly, Rbpjl is required for the differentiation of pancreatic adenoblasts, which regulate exocrine secretion[14,15], but may not affect endocrine secretion. In 12-week-old male Rbpjl-mutated mice, both trabecular and cortical bone volumes were reduced due to decreased osteoblasts and bone formation, whereas osteoclast number and bone resorption remained unchanged (Fig. 7a–e). Similar to

*Fam102a*–deficient cells, calvarial cells derived from Rbpjl-mutated mice showed reduced osteoblast differentiation in vitro (Fig. 7f). Gene expression analysis showed that *Sp7* expression decreased in Rbpjl-mutated cells, even though *Runx2* expression was unchanged (Fig. 7g), which is consistent with the results obtained in cells after Rbpjl knockdown. These results indicate that Rbpjl induces osteoblast differentiation through *Sp7* expression.

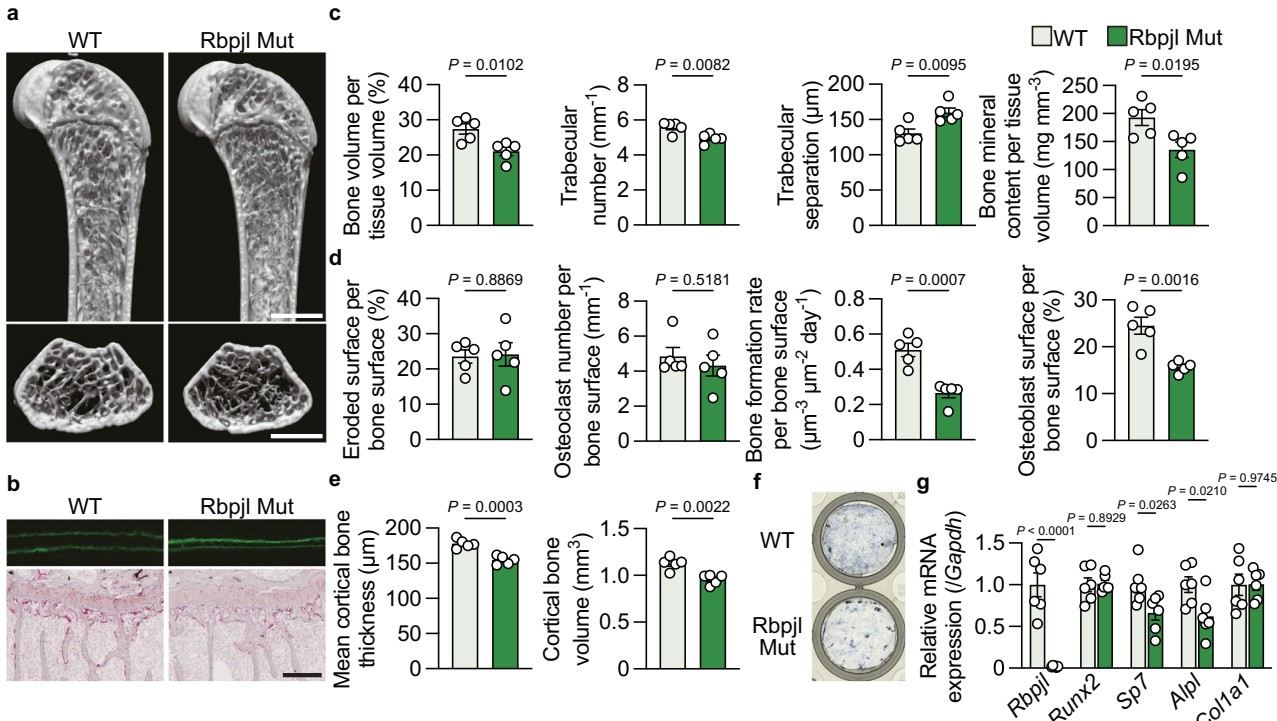

**Fig. 7 | Rbpjl-mutated mice exhibit an osteoporotic phenotype.**
**a** Representative μCT images of the distal femur in 12-week-old male Rbpjl-mutated mice and their littermate controls (scale bar, 1 mm). **b** New bone formation was determined by calcein double labeling (upper panel). TRAP staining of the proximal tibiae of 12-week-old male Rbpjl-mutated mice and their littermate controls (lower panel) (scale bar, 0.2 mm). **c** μCT analysis of the distal femur in 12-week-old male Rbpjl-mutated mice and their littermate controls (*n* = 5). **d** Bone morphometric analysis in the proximal tibiae in 12-week-old male Rbpjl-mutated mice and their littermate controls (*n* = 5). **e** Cortical bone analysis as measured by μCT in 12-week-old male Rbpjl-mutated mice and their littermate controls (*n* = 5). **f** ALP staining of calvarial cells derived from Rbpjl-mutated mice and their littermate controls cultured in osteogenic medium for 7 days. **g** mRNA expression of osteoblastic genes during differentiation in calvarial cells derived from Rbpjl-mutated mice and their littermate controls (*n* = 7; independent experimental replicates). WT, wild-type. Mut, mutated. Data are shown as the mean ± SEM. Statistical analyses were performed using unpaired two-sided Student's *t* test. Source data are provided as a Source Data file.

## Rbpjl and Runx2 cooperatively regulate *Sp7* expression, independently of Notch signaling

We examined the association between Rbpjl and Notch signaling to explore the mechanism by which Rbpjl regulates osteoblast differentiation. Although Rbpjl does not interact with the Notch intracellular domain (NICD) family or participate in Notch signaling[13], another study reported that Rbpjl inhibits Notch signaling through the substitution of Rbpj[37]. Therefore, we examined whether Rbpjl is involved in Notch signaling in osteoblasts. Rbpj interacted with NICD1, whereas negligible interaction was observed between Rbpjl and NICD1 (Supplementary Fig. 17a). NICD1 alone positively enhanced the promoter activity of hairy/enhancer-of-split related with YRPW motif 1 (Hey1), a downstream target of Notch signaling, whereas Rbpjl alone or with NICD1 had no effect (Supplementary Fig. 17b). In addition, Rbpjl-mutated osteoblastic cells showed no change in the expression of *Rbpj* and Notch signaling target genes (Supplementary Fig. 17c). As these results indicated that Rbpjl regulates osteoblast differentiation independently of Notch signaling, we examined whether Rbpjl regulates *Sp7* promoter activity. Because there are three putative Rbpjl-binding sites in the 2 kb upstream region of *Sp7* (Fig. 8a)[28], we examined Rbpjl binding to each site and found that Rbpjl binds to site 1 (Fig. 8b). However, the presence of Rbpjl did not affect the transcriptional activity of this region (Fig. 8c), suggesting that Rbpjl binds to the *Sp7* promoter but may not enhance promoter activity by itself. This was supported by the fact that overexpression of Rbpjl alone in MC3T3-E1 cells did not promote osteoblast differentiation (Supplementary Fig. 15a–c).

As Rbpjl has transcriptional activity in cooperation with pancreas specific transcription factor, 1a (Ptf1a), a transcription factor essential for the differentiation of pancreatic adenocytes[38], we considered the possibility that another factor is required for Rbpjl to regulate *Sp7* expression in osteoblasts. One of the transcription factors that positively regulates *Sp7* promoter activity in osteoblasts is Runx2 (Fig. 5c), and we thus hypothesized that Runx2 cooperates with Rbpjl to enhance *Sp7* expression. As expected, we demonstrated that the interaction between Rbpjl and Runx2 (Fig. 8d) and Rbpjl deficiency in MC3T3-E1 cells induced by CRISPR/Cas9-based gene editing (Supplementary Fig. 18a) suppressed osteoblast differentiation (Supplementary Fig. 18b, c). Furthermore, while the presence of Runx2 enhanced *Sp7* promoter activity in wild-type osteoblastic cells, this enhancement was significantly reduced in *Rbpjl*–deficient cells (Fig. 8e). In addition, overexpression of Fam102a only slightly enhanced Runx2-mediated *Sp7* promoter activity in *Rbpjl*–deficient osteoblastic cells compared to that in wild-type cells (Supplementary Fig. 18d). These results suggest that Rbpjl cooperates with Runx2 to regulate *Sp7* expression. We also examined the self-amplification ability of Rbpjl in osteoblasts and found that Rbpjl bound to its promoter region in MC3T3-E1 cells (Fig. 8f) and that the promoter activity of *Rbpjl* was enhanced after Runx2 overexpression (Fig. 8g and Supplementary Fig. 18e). Taken together, these results indicate that Rbpjl has a self-amplifying ability that is enhanced by Runx2 and increases its expression upon differentiation into osteoblasts.

## Discussion

Although total and healthy human life expectancy continues to increase in developed countries, the gap between total life expectancy and healthy life expectancy has not changed[39]. The number of years

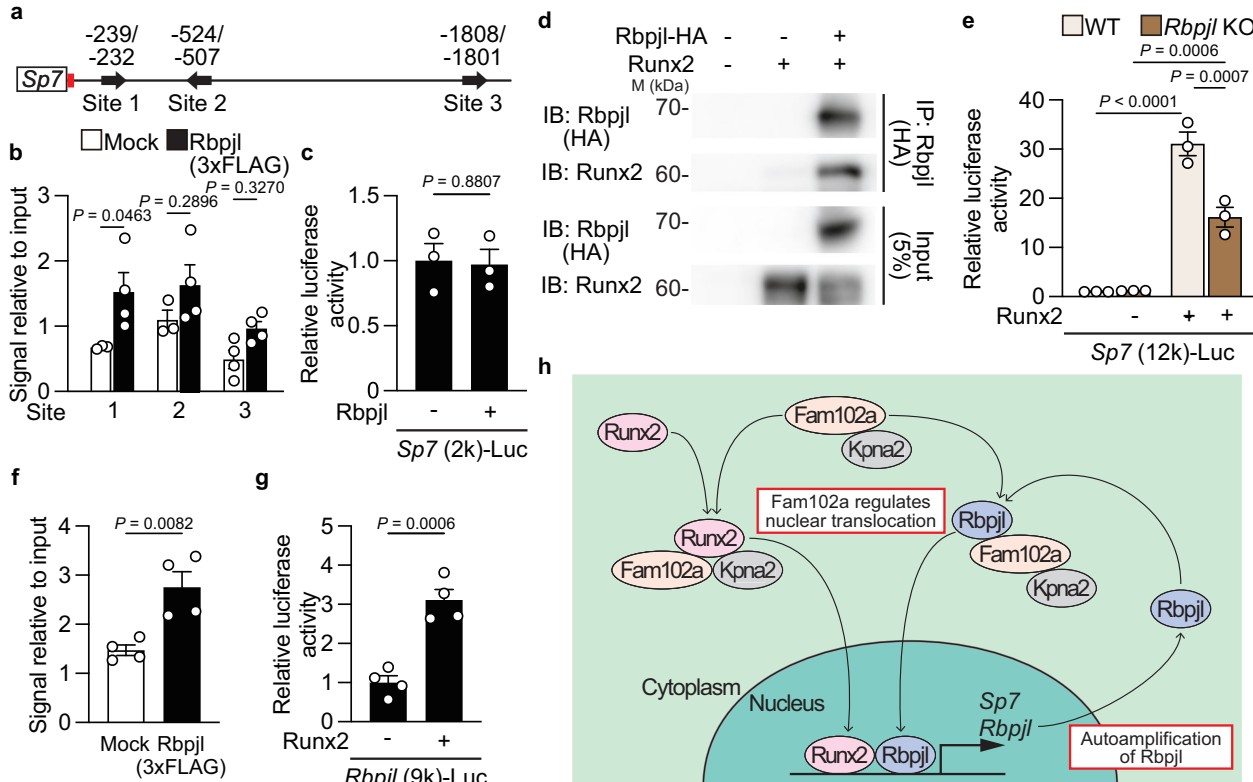

**Fig. 8 | Rbpjl and Runx2 cooperatively regulate *Sp7* expression independently of Notch signaling. a** The putative Rbpj/l-binding sites (black arrows) and a Runx2-binding site (red bar) in the mouse *Sp7* promoter regions. **b** The recruitment of Rbpjl to *Sp7* promoter in MC3T3-E1 cells stably expressing 3×FLAG-tagged Rbpjl examined by CUT&RUN assay (Site 1: Mock *n* = 3 and Rbpjl *n* = 4, Site 2: Mock *n* = 3 and Rbpjl *n* = 4, Site 3: *n* = 4 per group; independent experimental replicates). **c** The effect of Rbpjl expression on the reporter activity of *Sp7* (2 k)-Luc in MC3T3-E1 cells (*n* = 3; independent experimental replicates). **d** The analysis of the association between Rbpjl and Runx2 examined by Co-IP. Three independent experiments were performed. **e** The effect of Runx2 expression on the reporter activity of *Sp7* (12

k)-Luc in control and *Rbpjl*−deficient MC3T3-E1 cells (*n* = 3; independent experimental replicates) (WT, wild-type; KO, knockout). **f** The recruitment of Rbpjl to the *Rbpjl* promoter in MC3T3-E1 cells stably expressing 3×FLAG-tagged Rbpjl examined by CUT&RUN assay (*n* = 4; independent experimental replicates). **g** The effect of Runx2 expression on the reporter activity of *Rbpjl*-Luc in MC3T3-E1 cells (*n* = 4; independent experimental replicates). **h** Schematic view of osteoblast differentiation controlled by Fam102a, Rbpjl, and Runx2. M, molecular mass. Data are shown as the mean ± SEM. Statistical analyses were performed using unpaired two-sided Student's *t* test (**b**, **c**, **f**, **g**) or two-way ANOVA and Tukey's *post hoc* test (**e**). Source data are provided as a Source Data file.

lived with a disability refers to the period of life in which a person is in poor health and limited in normal activities of daily living, usually at the very end of life[40]. One of the major health challenges of modern age is to reduce the length of this debilitated period. This may be the foremost health challenge of the modern age. One of the major factors contributing to the increasing number of years lived with a disability is hip fracture due to bone fragility[41]. Therefore, understanding the mechanism of osteoporosis that leads to fragile bones and fractures is important for extending healthy life expectancy.

Osteoporosis is largely associated with the disruption of bone remodeling, which fundamentally controls dynamic bone homeostasis[1,2]. As osteoclasts and osteoblasts that play key roles in bone remodeling have different origins, few studies have been reported on the factors commonly involved in their respective differentiation. In this study, we identified Fam102a, a regulator of both osteoclast and osteoblast differentiation, as a bone-remodeling factor. Fam102a positively regulates both osteoclast and osteoblast differentiation; however, its function is osteoblast-dominant. Fam102a deficiency resulted in a phenotype resembling osteoporosis with low bone turnover, suggesting that it may be involved in senile osteoporosis, which is caused by a similar mechanism[42]. Fam102a contains an N-terminal C2 (NT-C2) domain, which is also present in EH domain-binding protein 1 (Ehbp1). Previous reports have shown that the NT-C2 domain of Ehbp1 functions as an adaptor that binds actin and

microfilaments and facilitates the attachment of microfilament-binding adaptors to membranes, which are potentially involved in intracellular trafficking[43]. However, the exact function of the NT-C2 domain remains unclear. Although importins have generally been shown to play an important role in the nuclear trafficking of intracellular proteins, previous studies have indicated that some transcription factors localize to the nucleus regardless of whether they interact directly with importins[44–48]. Fam102a may be involved in the nuclear translocation of these transcription factors that do not directly bind to importins and further analysis of the molecular mechanisms involved in Fam102a may lead to the elucidation of the regulatory mechanisms underlying the translocation of intracellular proteins to the nucleus.

We identified *Rbpjl*, a member of the *Rbpj* family, as a transcription factor regulated by Fam102a in the nucleus and cytoplasm. Rbpjl was unable to directly interact with importins (Supplementary Fig. 14b) and its nuclear translocation and self-amplification were regulated in a Fam102a–dependent manner. Moreover, we showed that Rbpjl positively regulates osteoblast differentiation by coordinating with Runx2 to modulate Osx expression. Although various factors are known to regulate Runx2 expression, including TGF-β, IGF, and Wnt signaling[7], only a few have been reported to regulate Osx expression, such as NFAT and p53[11,49]. Therefore, the function of Rbpjl as a regulator of Osx expression identified in this study represents a mechanism underlying osteoblast differentiation.

In summary, Fam102a regulates osteoblast differentiation by controlling the nuclear translocation of transcription factors in osteoblasts, and Rbpjl, which functions downstream of Fam102a, is also involved in osteoblast differentiation (Fig. 8h). Thus, the Fam102a-Rbpjl axis plays an important role in osteoblasts, providing insights into bone remodeling.

## Methods

### Mice

Ctsk[Cre/+] mice were kindly provided by Dr. T. Nakamura[16]. Nfatc1-floxed mice were kindly provided by Dr. A. Rao[18]. Actb-Cre and Sp7-Cre mice were generated as previously described[50,51]. Fam102a-floxed mice were generated as follows; a targeting vector was designed to insert a single loxP site upstream of exon 2, and a loxP-flanked neomycin resistance cassette downstream of exon 3 of the Fam102a gene (Supplementary Fig. 6a–c). Rbpjl-mutated mice were generated by i-GONAD method as described below. Wild-type mice (C57BL/6J) were obtained from CLEA Japan. All of the mice were maintained under specific pathogen-free conditions. Mice were maintained in a temperature and humidity-controlled room on a 12-hour light cycle with ad libitum access to a water and standard laboratory chow diet. All animal experiments were approved by the Institutional Animal Care and Use Committee in Institute of Science Tokyo (A2023-026C4).

### i-GONAD

i-GONAD was conducted according to a published protocol[36]. In brief, a CRISPR RNA (crRNA) (GGGAGCGCAGGCGATTGAAA, targeted to exon 7 of the Rbpjl genomic locus of Mus Musculus), trans-activating crRNA (tracrRNA), and HiFi Cas9 Nuclease were purchased from Integrated DNA Technologies (tracrRNA, Cat#1072532; HiFi Cas9 Nuclease, Cat#1081060). The crRNA-tracrRNA-Cas9 (RNP) complex was injected into the oviduct lumen of a female wild-type mouse which had been mated on the previous day, due to efficient editing genome at 0.7 dpc. Immediately after injection, the Kimwipe towel-covered oviduct was sandwiched between tweezer-type electrodes. Electroporation was performed using a BEX (model no. CUY21EditII) with the following parameters (square (mA), (+/−), Pd V: 80 V, Pd A: 150 mA, Pd on: 5.00 ms, Pd off: 50 ms, Pd N: 3, decay: 10%, decay type: Log). The offspring from the i-GONAD−treated females were analyzed for genotype using a target-specific primer pair (Forward: 5′-ACATCAATACCCAGATACCCGAC-3′, Reverse: 5′-CCAGGCAGCTCTATTCTTTGAT-3′). The selected mutated mice were backcrossed for three generations before they were used in the experiments.

### Genechip analysis

Wild-type osteoclasts derived from bone marrow and Ctsk[Cre/+] Nfatc1[flox/flox] osteoclasts derived from splenic cells were induced by RANKL stimulation, as described below. Gene expression profiling data from wild-type and Runx2[−/−] osteoblasts was kindly provided by Dr. K. Nishikawa[52]. The total RNA extracted from these cells was used for cDNA synthesis of biotinylated cRNA through in vitro transcription. After cRNA fragmentation, hybridization with the Mouse Genome 430 2.0 Array (Affymetrix) was performed.

### Cell culture

For in vitro osteoclast differentiation, primary bone marrow cells (1 × 10⁵ cells per cm²) or primary splenic cells (1 × 10⁶ cells per cm²) were maintained in α-Minimum Essential Medium (α-MEM) supplemented with 10% fetal bovine serum (FBS), 100 U ml⁻¹ penicillin, 100 μg ml⁻¹ streptomycin, and 10 ng ml⁻¹ of M-CSF (R&D systems, Cat#216-MC-500) for 2 days to obtain macrophages. Then the macrophages obtained were cultured in the above medium supplemented with 50 ng ml⁻¹ of RANKL (PeproTech, Cat#315−11) for 3 days. The culture medium was changed every second day. Osteoclastogenesis was evaluated by TRAP

staining. TRAP-positive multinucleated cells (more than 3 nuclei) were counted. For in vitro osteoblast differentiation, calvarial cells were isolated from the calvarial bone of newborn mice (P0–P3) by enzymatic digestion in α-MEM with 0.1% collagenase and 0.2% dispase, and maintained with α-MEM supplemented with 10% FBS, 100 U ml⁻¹ penicillin, and 100 μg ml⁻¹ streptomycin. After proliferating to subconfluency, these cells were reseeded (1 × 10⁴ cells per cm²) and maintained for 3 days. Then the obtained mesenchymal cells were cultured in an osteogenic medium (100 mM ascorbic acid (Nacalai Tesque, Cat#13570−82), 5 mM β-glycerophosphate (Sigma-Aldrich, Cat#G9422), and 10 mM dexamethasone (Wako, Cat#041−18861)). The culture medium was changed every third day. After 7 days, alkaline phosphatase staining was performed, and after 21 days, bone nodule formation was assessed by Alizarin Red S staining. β-estradiol (Tocris Bioscience, Cat#2824) was added to the primary osteoclast precursors (24 h after the first RANKL stimulation) and osteoblast precursors using MC3T3-E1 cells (Riken BRC, RBRC-RCB1126, Lot. 057) (7 days after cultured in osteogenic medium). INCA-6 (Tocris Bioscience, Cat#2162) was added to osteoblast precursor cells using MC3T3-E1 cells (7 days after being cultured in osteogenic medium). FGF2 (Pepro Tech, Cat#450−33) (10 ng ml⁻¹) was added to the osteoblast precursors using MC3T3-E1 cells to assess MAPK signaling.

### CRISPR/Cas9

A crRNA (GGGAGCGCAGGCGATTGAAA, targeted to exon 7 of the Rbpjl genomic locus from the Mus Musculus), tracrRNA (Cat#1072532), and espCas9 (Cat#1081060) were purchased from Sigma-Aldrich. MC3T3-E1 cells were transfected with the RNP complex and Rbpjl knockout cells were isolated[53]. Rbpjl knockout cells were analyzed for genotyping using the above primer pair.

### Cell proliferation assay

The 3-[4,5-dimethylthiazol-2-yl]-2,5-diphenyltetrazolium bromide (MTT) assay was performed to determine cell proliferation. Calvarial cells were plated at a density of 1 × 10⁴ cells per well in 96-well plates. After incubation in α-MEM for 24 hr, the cells were incubated for 1 hr with an MTT reagent (Nacalai Tesque, Cat#23547−76) at each time point. DMSO was then used to dissolve the formazan crystal. The absorbance was recorded at 490 nm using a microplate reader.

### Bone analysis

Bone analysis was conducted using 12-week-old mice, except for the 3-week-old Ctsk[Cre/+] Nfatc1-floxed male mice. The right femur and lumbar vertebrae were fixed in 70% ethanol for μCT analysis. μCT scanning was performed using a Scan Xmate-A100S Scanner (Comscantechno). Three-dimensional microstructural images were reconstructed, and then structural indices were calculated using TRI/3D-BON software (v10.01.37.47-H-64; RATOC System Engineering) in accordance with the American Society for Bone and Mineral Research (ASBMR) guidelines for μCT analysis of rodent bone microstructure[54]. The right tibia was subjected to histological analysis. The undecalcified tibiae were embedded in glycol methacrylate and stained with toluidine blue and TRAP[50,55,56]. The histomorphometric parameters were calculated based on the ASBMR guidelines for bone histomorphometry[57].

### Histological analysis

The organs were fixed in 4% paraformaldehyde/phosphate-buffered saline (Nacalai Tesque, Cat#09154−85) for approximately 12 to 24 hr, and subsequently embedded in paraffin after a series of dehydration steps. The tissues were sectioned at 4 μm thickness, deparaffinized using Fast Solve (FALMA), and rehydrated through a graded ethanol series followed by Milli-Q water (Merck Millipore)[58]. The sections were then stained with hematoxylin (Muto Pure Chemicals) and eosin (Nacalai Tesque), dehydrated through the ethanol series and Fast Solve, and finally mounted using Multi Mount 480 (Matsunami Glass).

All images were obtained using the BZ-X700 all-in-one fluorescence microscope (v1.4.0.1; KEYENCE).

## Quantitative PCR

Total RNA was isolated using Sepasol-RNA I Super G (Nacalai Tesque, Cat#09379–97) according to the manufacturer's instructions. cDNA was synthesized from total RNA with ReverTra Ace qPCR RT Master Mix along with gDNA Remover (Toyobo, Cat#FSQ-301). qPCR analysis was performed with the CFX384 Touch Real-Time PCR Detection System (v3.11517.0823; Bio-Rad Laboratories) using THUNDERBIRD SYBR qPCR Mix (Toyobo, Cat#QPS201). The expression of genes was normalized to *Gapdh*. Expression levels for all the genes were analyzed using the standard curve method as previously described[55]. The sequences of the primers are shown in the Supplementary Table 2.

## RNA sequencing analysis

Total RNA was extracted with Sepasol-RNA I Super G (Nacalai Tesque, Cat#09379–97). Data were acquired on an Ion Proton sequencer (Thermo Fisher Scientific) and analyzed using the CLC genomic Workbench (v12.0.3; CLC bio) according to the manufacturer's instructions. Functional analysis was performed on DEGs with DAVID 6.8[22] and Quick GO[23]. The datasets generated in the present study are available in the Gene Expression Omnibus (GEO) database (https://www.ncbi.nlm.nih.gov/geo/) under accession no. GSE211533.

## Western blotting

Cell lysate of calvarial cells or MC3T3-E1 cells was fractionated by Sodium dodecyl sulfate-polyacrylamide gel electrophoresis (SDS-PAGE) and transferred to a polyvinylidene difluoride (PVDF) membrane. After incubation with a blocking buffer, the proteins were subjected to western blot analysis using the specific antibodies for HA (TANA2, MBL, Cat#M180-3), Myc (My3, MBL, Cat#M192–3), LaminB1 (A −11, Santa Cruz Biotechnology, Cat#sc-377000), α-tubulin (DM1A, Santa Cruz Biotechnology, Cat#sc-32293), β-actin (AC−15, Sigma-Aldrich, Cat#A5441), FLAG (M2, Sigma-Aldrich, Cat#F1804), Runx2 (D1L7F, Cell Signaling Technology, Cat#12556), phosphor-ERK (D13.14.4E, Cell Signaling Technology, Cat#4370), ERK (137F5, Cell Signaling Technology, Cat# 4695), phosphor-p38 (D3F9, Cell Signaling Technology, Cat#4511), p38 (D13E1, Cell Signaling Technology, Cat#8690), phosphor-JNK (81E11, Cell Signaling Technology, Cat#4668) and JNK (N/A, Cell Signaling Technology, Cat#9252). The rabbit anti-Rbpjl polyclonal antibody was prepared by Sigma-Aldrich against SALPRLPNAQEPAPDADTL, a sequence near the C terminus of mouse Rbpjl, and affinity purified using the synthetic peptide. Nuclear proteins were prepared with a nuclear extract kit in accordance with the manufacturer's protocol (Active Motif, Cat#40010).

## Co-Immunoprecipitation analyses

Human Embryonic Kidney (HEK) 293 cells (Riken BRC, RBRC-RCB1637, Lot. 024) were transfected with each expression gene, solubilized in lysis buffer (Thermo Fisher Scientific, Cat#87788) and supplemented with a complete protease inhibitor cocktail (Roche Applied Science, Cat#11836170001). Immunoprecipitation was performed by incubation with an anti-DDDDK-tag bearing mAb-Magnetic Beads (FLA−1GS, MBL, Cat#M185−11), anti-HA-tag mAb-Magnetic Beads (TANA2, MBL, Cat#M180−11), or anti-Myc-tag mAb-Magnetic Beads (PL14, MBL, Cat#M047−11) in accordance with the manufacturer's protocol. Immune complexes were separated by electrophoresis followed by blotting with anti-FLAG M2 (M2, Sigma-Aldrich, Cat#F1804), anti-HA (TANA2, MBL, Cat#M180−3), and anti-Myc (My3, MBL, Cat#M192−3) antibodies.

## Immunocytochemical staining

Calvarial cells were seeded on collagen-type I-P coated glass bottom dishes (Thermo Fisher Scientific). Cells were blocked with 5% FBS for 30 min at room temperature, followed by staining with an anti-Rbpjl or anti-Runx2 antibody (D1L7F, Cell Signaling Technology, Cat#12556) in 5% FBS overnight at 4 °C. After washing with PBS-T, cells were incubated with Alexa 488-labeled anti-rabbit IgG antibody (N/A, Thermo Fisher Scientific, Cat#A21206) and 1 μg ml$^{-1}$ Hoechst 33342 (Thermo Fisher Scientific, Cat#H1399) for 1 hr at room temperature.

## Retoroviral and Lentiviral gene transfer

The retroviral vectors pMXs-Runx2-IRES-GFP, pMXs-Fam102a-IRES-GFP, and pMXs-Rbpjl-IRES-GFP were constructed by inserting DNA fragments encoding Runx2, Fam102a, and Rbpjl into a pMXs-IRES-GFP vector. The retrovirus supernatants were obtained by transfecting the retroviral vectors into the Plat-E packaging cell line using FuGENE HD (Promega, Cat#E2311). The lentiviral vector pLVSIN-Rbpjl-HA-puro and pLVSIN-Rbpjl-3xFLAG-puro were constructed by inserting DNA fragments encoding Rbpjl, HA, and 3xFLAG into a pLVSIN-EF1α-puro vector. Lentiviral shRNA expression vectors for Fam102a (SHCLNG-NM_153560), Rbpjl (SHCLNG-NM_009036), and TurboGFP positive control vector (SHC003) were purchased from Sigma-Aldrich. The lentivirus vectors were co-transfected with MISSION Lentiviral Packaging Mix (Sigma-Aldrich, Cat#SHP001) into HEK 293 T cells (Riken BRC, RBRC-RCB2202, Lot. 021). To generate Rbpjl-HA and Rbpjl-3xFLAG stably expressed cell lines, MC3T3-E1 cells were transfected with expression vectors and isolated by limiting dilution with 5 μg ml$^{-1}$ puromycin. The isolated Rbpjl-HA and Rbpjl-3xFLAG stably expressed cells were analyzed by western blotting using specific antibodies.

## CUT&RUN assay

CUT&RUN assay was performed using the CUT&RUN Assay Kit (Cell Signaling Technology, Cat#86652) according to the manufacturer's instructions[59]. Briefly, 100,000 cells for each reaction and input sample were bound to activated Concanavalin A-coated magnetic beads and permeabilized. The bead-cell complexes were incubated with anti-HA, anti-rabbit IgG (N/A, Cell Signaling Technology, Cat#66362), or anti-FLAG M2 (M2, Sigma-Aldrich, Cat#F1804) antibodies at 4 °C for 2 hr. Cells were resuspended with pAG-MNase Enzyme and incubated at room temperature for 1 hr. DNA fragments were purified using DNA Purification Buffers and Spin Columns (Cell Signaling Technology, Cat#14209). The sequences of the primers are shown in the Supplementary Table 3.

## Reporter gene assay

The reporter plasmids of *Rbpjl*-Luc, *Sp7* (12 k)-Luc, and *Sp7* (2 k)-Luc were constructed by subcloning from BAC clones (B6Ng01-148B24 and B6Ng01-144H14) provided by the RIKEN BRC through the National BioResource Project of MEXT, Japan. The reporter plasmid of *Fam102a*-Luc was constructed by cloning from the mouse genomic DNA. The reporter plasmid of *Sp7* (12 k) mutated-Luc was constructed from *Sp7* (12 k)-Luc vector using the In-Fusion kit (Takara, Cat#638909). The reporter plasmids and the expression plasmids were transfected into MC3T3-E1 cells or NIH3T3 cells (Riken BRC, RBRC-RCB1862, Lot. 011) using X-tremeGENE HP DNA Transfection Reagent (Roche, Cat#6366244001) or FuGENE HD (Promega, Cat#E2311). After 48 hr, a dual luciferase assay (Promega, Cat#E1910) was performed according to the manufacturer's protocol.

## Statistical analysis

All data are expressed as the mean ± SEM of at least three independent experiments. Two data sets were compared using unpaired two-sided Student's *t* test. Multiple comparisons were performed using one- or two-way ANOVA followed by Tukey's or Sidak's *post hoc* test. A *P* value of <0.05 was considered statistically significant. All statistical analyses were performed using Prism version 8 (GraphPad Software).

## Reporting summary

Further information on research design is available in the Nature Portfolio Reporting Summary linked to this article.

## Data availability

The sequencing data generated in this study have been deposited in the GEO database under accession code GSE211533. Source data are provided in this paper. All other data supporting the findings of this study are available from the corresponding author upon reasonable request. Source data are provided with this paper.

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

## Acknowledgements

We thank K. Yusoon, M. Inoue, R. Denda and T. Gondo for helpful discussions and technical assistance. This work was supported in part by Advanced Research and Development Programs for Medical Innovation under JP20gm0810003 (T.N.) and JP22gm6110027 (M.H.) from Japan Agency for Medical Research and Development (AMED); Grant-in-Aid for Scientific Research (A) (T.N.), Scientific Research (B) (M.H.), and Challenging Research (Exploratory) (M.H. and T.N.) from the Japan Society for the Promotion of Science (JSPS); and grants from Takeda Science Foundation (M.H. and T.N.), Astellas Foundation for Research on Metabolic Disorders (T.N.), Daiichi Sankyo Foundation (T.N.), and Secom Science and Technology Foundation (T.N.).

## Author contributions

Y.Y., A.L., M.H., H.T., and T.N. conceived and designed the research. Y.Y., A.L., F.S., Y.T., and M.H. performed the research and acquired the data. Y.Y., M.H., and T.N. analyzed and interpreted the data. Y.Y. wrote the manuscript with support from M.H., A.L., H.T., M.S., and T.N. All authors were involved in drafting and revising the manuscript.

## Competing interests

The authors declare no competing interests.
