## [Transparent Peer Review file · Nature Communications]

Fam102a translocates Runx2 and Rbpjl to facilitate Osterix expression and bone formation

Corresponding Author: Dr Mikihiro Hayashi

Version 0:

Reviewer comments:

Reviewer #1

(Remarks to the Author)

In this manuscript, Yamashita and colleagues report Fam102a and Rbpjl as new Runx2-binding partners that regulate Runx2 function during osteoblast differentiation. Strengths of this manuscript include use of (1) multiple new mouse models demonstrating similar states of low bone turnover osteopenia, (2) robust transcriptomic datasets to identify Fam102a, and (3) description of a role for Fam102a in regulating nuclear trafficking of key transcription factors involved in osteoblast maturation. Given the important role of Runx2 as a master regulator transcription factor that drives osteoblast differentiation, these data may be of interest to the bone biology research community. Despite these strengths, enthusiasm for the manuscript is limited to a moderate/large degree by (1) lack of solid mechanistic insight into how Fam102a works to promote Runx2 activity, (2) unclear rationale for studying Rbpjl, and (3) major shortcomings regarding controls for co-immunoprecipitation studies and inadequate attention to sex as a biologic variable for the bone phenotypes reported.

Major points:

1. In Figure 5, data regarding the Runx2/Fam102a/Kpna2 interaction is under-developed. Ideally the authors would extend this molecular mechanism by mapping regions of each protein responsible for the interaction and testing interaction-deficient mutants. In general, the reader comes away from the manuscript without a solid grasp of what Fam102a is doing to regulate Runx2 localization in osteoblasts. Furthermore, how does Fam102a interact with the large of other binding partners that have been reported to control Runx2 activity at multiple levels over the past ~20 years?
2. Based on the results showing that Fam102a controls Runx2 activity, it is unclear why the authors needed to evoke another Fam102a-binding partner (Rbpjl) to explain the osteopenia phenotype observed. Does Runx2 overexpression rescue the phenotype of the Fam102a mutant cells or mice?
3. Critical controls are lacking to interpret the co-immunoprecipitation studies performed in Figures 5d, 5i, 6c, and 8d. For all co-IP studies, the authors must show that the 'prey' protein does not bind non-specifically to the beads that are used in the absence of the over-expressed 'bait' target protein. Ideally, co-IP studies would be performed using endogenous proteins where possible in order to avoid potential over-expression artifacts.
4. The authors must indicate the age and sex of the mice studied for microCT and histomorphometry analysis in the methods, results, and figure legends. Consideration of sex as a biologic variable is absolutely critical for manuscript detailing skeletal phenotypes.

Minor points:

1. It would be helpful to introduce Fam102a a bit more- what kind of protein is it? What domains are present? Has it been previously reported to interact with the nuclear import machinery?
2. Figure 1g: ideally they would mutate the Runx2 binding site in this enhancer construct
3. Figure 1h: ideally the localization of endogenous Fam102a would be studied
4. Figure 2: use of Ctsk-Cre is problematic since this Cre is also active in some mesenchymal osteoblast precursors on periosteal bone surfaces. Therefore, the authors should measure periosteal bone formation.

5. Figure 3: in addition to trabecular bone microCT, the authors should report cortical bone parameters
6. Figure 4: are there phenotypes in the global Fam102a null mice outside of bone?
7. Figure 5g: quantification should be performed for the Runx2 immunofluorescence data
8. Figure 6: it is unclear how RNA-seq helps to identify direct 'clients' of Fam102a, and why it is necessary to study factors other than Runx2 to explain the phenotype observed here
9. Figure 8: as noted above, these results are interesting, but the relationship to Fam102a should be clarified
10. Overall, the manuscript would benefit to some degree from editing for grammar/syntax

Reviewer #2

(Remarks to the Author)

The authors provide a comprehensive study regarding the function of Fam102a in bone remodeling cell types. They essentially confirm previous data regarding the role of this molecule in osteoclast differentiation, which explains, together with the low bone mass phenotype of mice ubiquitously lacking Fam102a, why the authors focused on the function of Fam102a in osteoblasts. They show that Fam102a is involved in the nuclear translocation of key transcription factors (Runx2, but also Rbpjl), which eventually enhances the expression of Osterix (encoded by the Sp7 gene) to promote bone formation. Overall, this is a convincingly performed and presented study. However, the authors should be much more careful with their statements about the potential clinical relevance. In fact, they should focus less on osteoporosis as a major health problem, but rather discuss what is known about functions of Fam102a (also known as Eeig1), not only in osteoclastogenesis (Refs 16,17), but also in other cell types.

Specific comments:

- 1) One major goal of the authors was to identify a molecule required for differentiation of both, osteoclasts and osteoblasts. Since it is quite obvious, however, that these two cell types are entirely different in terms of progenitor cells, morphology, mode of action and regulatory molecules, the relevance of such a research question is at least debatable. Moreover, whereas the authors state twice that there are only few reports identifying factors required for the differentiation of both bone remodeling cell types, they don't include the respective references. It would be very informative to know, which examples the authors are referring to.
- 2) Especially given the striking difference in TRAP staining shown in Supplementary Figure 1c, it would be advantageous to display the expression values for known markers of osteoclastogenesis and osteoblastogenesis as obtained by the Gene Chip analyses shown in Fig. 1a. Similarly, the authors should provide expression levels for known osteoblast markers as obtained by the RNA sequencing experiment shown in Fig. 5a.
- 3) Since a mouse deficiency model for Fam102a/Eeig1 has been described previously, it would be important to better illustrate the targeting strategy used by the authors for the present study. It is also not specified, how exactly the global deletion was achieved.
- 4) It would be informative if Fig. 8a could also display Runx2 binding sites in the Sp7 promoter.
- 5) The statement that "elucidation of the Fam102a function may lead to new options for the treatment of age-related osteoporosis" needs to be supported by additional information. How should such a treatment be pursued, if Fam102a is located intracellularly? Are there known agonists for Fam102a or proteins with similar function? In which tissues and cell types is Fam102a expressed? Are there any other phenotypes, for instance reduced pancreas weight, in mice ubiquitously lacking Fam102a? And finally, should a molecule promoting the differentiation of both, osteoclasts and osteoblasts, be considered as a drug target to counteract osteoporosis? In any case, as there is no human genetic evidence so far to support a key role of Fam102a in bone remodeling, the clinical relevance of mouse and cell culture data should not be over-estimated.

Version 1:

Reviewer comments:

Reviewer #1

(Remarks to the Author)

All previous comments have been addressed in a satisfactory manner. One minor point: the immunoblot in panel 6c appears to be cropped between lanes 3 and 4. The figure should be modified accordingly.

Reviewer #2

(Remarks to the Author)

The authors have adequately addressed all remaining issues and further improved their manuscript.

Responses to reviewers' comments on NCOMMS-24-10523

We are grateful to the two reviewers for their positive comments and invaluable suggestions. To address all of the issues raised by the reviewers, we have incorporated new data after performing additional experiments, and carefully revised the manuscript accordingly. We believe that the revised manuscript has taken into consideration essentially all of the comments, and hope that it has been improved to the satisfaction of the editors and reviewers. Please find our point-by-point response to each comment by the reviewers below.

Reviewer #1:

In this manuscript, Yamashita and colleagues report Fam102a and Rbpjl as new Runx2-binding partners that regulate Runx2 function during osteoblast differentiation. Strengths of this manuscript include use of (1) multiple new mouse models demonstrating similar states of low bone turnover osteopenia, (2) robust transcriptomic datasets to identify Fam102a, and (3) description of a role for Fam102a in regulating nuclear trafficking of key transcription factors involved in osteoblast maturation. Given the important role of Runx2 as a master regulator transcription factor that drives osteoblast differentiation, these data may be of interest to the bone biology research community. Despite these strengths, enthusiasm for the manuscript is limited to a moderate/large degree by (1) lack of solid mechanistic insight into how Fam102a works to promote Runx2 activity, (2) unclear rationale for studying Rbpjl, and (3) major shortcomings regarding controls for co-immunoprecipitation studies and inadequate attention to sex as a biologic variable for the bone phenotypes reported.

We would like to thank the reviewer #1 for the helpful and invaluable comments. Please see below for our responses to each of the comments that were raised. We hope that the revisions we have made and the new data we have included are satisfactory.

Major points:

1. In Figure 5, data regarding the Runx2/Fam102a/Kpna2 interaction is under-developed. Ideally the authors would extend this molecular mechanism by mapping regions of each protein responsible for the interaction and testing interaction-deficient mutants. In general, the reader comes away from the manuscript without a solid grasp of what Fam102a is doing to regulate Runx2 localization in osteoblasts. Furthermore, how does Fam102a interact with the large of other binding partners

that have been reported to control Runx2 activity at multiple levels over the past ~20 years?

We appreciate the reviewer's positive and insightful comments. As pointed out by the reviewer #1 in the major point #3, we have repeated all co-immunoprecipitation (Co-IP) experiments using appropriate negative controls. Most of the results were consistent with our previous results, with the exception of the direct binding of Runx2-Fam102a. We found that Kpna2 is required for the interaction between Runx2 and Fam102a, while this finding does not alter the conclusions of our manuscript (see details in the response to the Comment #3 by the Reviewer #1). Based on this result, we have revised Fig. 5 and the corresponding text. Since Fam102a and Runx2 do not bind directly, we focused on the interaction between Fam102a and Kpna2 and generated mutants of both Fam102a and Kpna2 to find the binding regions. We revealed that Fam102a binds to the armadillo repeat (Arm) region of Kpna2, and that both the N- and C-terminal of Fam102a are necessary for this interaction. These findings further strengthen the mechanism of the Runx2/Fam102a/Kpna2 interaction. We have included these points in the revised manuscript (Fig. 5i; Supplementary Fig. 13a,b; page 17, lines 252–257).

Not many factors are known to directly bind Runx2 to modulate its transcriptional activity; Yap1, Taz, Hes1, and Stat1 are prominent examples (Zaidi SK, et al., EMBO J. 2004; Cui CB, et al., Mol Cell Biol. 2003; McLarren KW, et al., J Biol Chem. 2000; Kim S, et al., Genes Dev. 2003). We examined whether Fam102a interacts with these Runx2 binding partners, and found that these molecules lack the ability to interact directly with Fam102a, using Fam102a-Kpna2 binding as a positive control. Therefore, these results suggest that these known Runx2 binding factors may not be involved in the mechanism by which Fam102a regulates Runx2 activity during osteoblast differentiation. We have included these new results in the revised manuscript (Supplementary Fig. 13c; pages 17–18, lines 257–262).

2. Based on the results showing that Fam102a controls Runx2 activity, it is unclear why the authors needed to evoke another Fam102a-binding partner (Rbpjl) to explain the osteopenia phenotype observed. Does Runx2 overexpression rescue the phenotype of the Fam102a mutant cells or mice?

We have shown that nuclear expression of Runx2 was significantly, but incompletely (approximately 75%), reduced in *Fam102a*-deficient osteoblasts (Fig. 5e). Despite the partial reduction in nuclear localization of Runx2, *Fam102a*-null mice developed significant bone loss. Therefore, we hypothesized that Fam102a may regulate osteoblast differentiation through a mechanism other than Runx2 nuclear localization. To explore

potential alternative regulators, we examined transcription factors downstream of Fam102a and revealed that Fam102a regulates Runx2 activity through Rbpjl. In revised manuscript, we have clearly stated this point (page 18, lines 267–271).

Following the reviewer's suggestion, we examined the effect of overexpression of Runx2 in *Fam102a*-deficient osteoblasts. The impaired osteoblast differentiation upon loss of *Fam102a* was rescued by Runx2 overexpression, suggesting that Runx2 functions downstream of Fam102a. We believe that this supports our conclusions and have described the new results in Supplementary Fig. 12c,d and on page 16, lines 240–242 in the revised manuscript.

3. Critical controls are lacking to interpret the co-immunoprecipitation studies performed in Figures 5d, 5i, 6c, and 8d. For all co-IP studies, the authors must show that the 'prey' protein does not bind non-specifically to the beads that are used in the absence of the over-expressed 'bait' target protein. Ideally, co-IP studies would be performed using endogenous proteins where possible in order to avoid potential over-expression artifacts.

We appreciate the reviewer raising this concern. As suggested by the reviewer, we have repeated all Co-IP experiments using samples expressing the 'prey' protein without expressing the 'bait' protein as a negative control. The results of all the Co-IP experiments were consistent with our previous findings, with the exception of the direct binding of Runx2-Fam102a. Although we were unable to detect the direct binding of Runx2-Fam102a, the loss of Fam102a reduces the nuclear expression of Runx2 (Fig. 5e and 5f), and Fam102a binds to Kpna2 directly (Fig. 5h). These results led us to hypothesize that Kpna2 is necessary for the interaction between Fam102a and Runx2. By co-expressing Fam102a, Runx2, and Kpna2, we showed that Fam102a and Runx2 bind indirectly (Fig. 5i). These findings suggest that the Fam102a/Kpna2/Runx2 interaction is necessary for Fam102a to regulate osteoblast differentiation. We have included these new results in the revised manuscript (Fig. 5i; page 17, lines 252–253). We also agree with the reviewer that it could be informative to assess the expression and interactions of endogenous proteins. However, due to the lack of suitable antibodies for endogenous Fam102a or Rbpjl, we performed the Co-IP experiments using overexpressing cells. We hope that the reviewer finds this approach acceptable.

4. The authors must indicate the age and sex of the mice studied for microCT and histomorphometry analysis in the methods, results, and figure legends. Consideration of sex as a biologic variable is absolutely critical for manuscript

detailing skeletal phenotypes.

We appreciate the reviewer's insightful comments. As suggested, we have added corresponding descriptions to the Methods, Results, and Figure legends. In addition, we analyzed *Fam102a*-deficient female mice and found a reduction in femoral bone mass, which is consistent with the findings in male mice. We have included these new results in the revised manuscript (Supplementary Fig. 9a; page 13, lines 189–190).

Minor points:

1. It would be helpful to introduce Fam102a a bit more- what kind of protein is it? What domains are present? Has it been previously reported to interact with the nuclear import machinery?

We apologize for the lack of description of what is known about Fam102a. Fam102a has been reported to contain an N-terminal C2 (NT-C2) domain, which is also found in the Ehbp1 protein. Previous reports have shown that the NT-C2 domain of Ehbp1 functions as an adaptor that binds actin and microfilaments, and facilitates the attachment of microfilament-binding adaptors to the membrane, suggesting that the NT-C2 domain may be involved in intracellular trafficking. However, the exact function of the NT-C2 domain remains unclear, as is its role in Fam102a. In the revised manuscript, we have added the description in the main text (page 26, lines 382–386).

2. Figure 1g: ideally they would mutate the Runx2 binding site in this enhancer construct

Following the reviewer's suggestion, we generated a vector lacking the Runx2 binding region within the enhancer region of Fam102a and examined the ability of Runx2 to activate Fam102a transcription. Our results showed that Runx2 directly binds to the enhancer region of Fam102a and upregulates its reporter signal, supporting our conclusion. We have included this new result in the revised manuscript (Supplementary Fig. 5d,f; page 10, lines 139–140).

3. Figure 1h: ideally the localization of endogenous Fam102a would be studied

We tested commercially available antibodies that has been used in previous reports. However, western blotting did not show clear bands in the expected molecular weight range, and cellular immunostaining showed only nonspecific results (Reviewer-only Figure). We agree with the reviewer that analysis of endogenous Fam102a is important, but we are unable to perform additional experiments due to the lack of antibodies capable of analyzing endogenous Fam102a. We hope that the reviewer will understand this

limitation.

Reviewer-only Figure: The lack of antibodies with sufficient capacity to analyze endogenous Fam102a.

a, Fam102a localization in primary osteoblasts. Nuclei were stained with Hoechst 33342 (blue). **b**, Representative western blot images of Fam102a in *Fam102a^{+/+}* and *Fam102a^{D/D}* calvarial cells cultured in day0, 7, and 21.

4. Figure 2: use of Ctsk-Cre is problematic since this Cre is also active in some mesenchymal osteoblast precursors on periosteal bone surfaces. Therefore, the authors should measure periosteal bone formation.

As the reviewer pointed out, previous reports have shown that *Ctsk^{Cre}* is also expressed in periosteal skeletal stem cells, which contribute to cortical bone homeostasis (Debnath S, et al., Nature. 2019; Yang W, et al., Nature. 2013; Han Y, et al., J. Clin. Invest. 2019). To assess the impact of Fam102a deficiency in *Ctsk⁺* cells on cortical bone, we examined the cortical bone parameters of the distal femur using μ CT. In comparison to control mice, *Ctsk^{Cre/+} Fam102a^{flx/ Δ}* mice had similar cortical bone thickness, volume, and periosteal perimeter, suggesting that *Fam102a* deletion in *Ctsk⁺* cells do not affect periosteal bone formation. We have added these new results in the revised manuscript (Supplementary Fig. 7; page 11, lines 158–162).

5. Figure 3: in addition to trabecular bone microCT, the authors should report cortical bone parameters

Following the reviewer's suggestion, we examined the cortical bone parameters of the distal femur in *Sp7-Cre⁺ Fam102a^{flx/ Δ}* mice. μ CT analysis revealed that cortical bone thickness and volume were significantly reduced in male *Sp7-Cre⁺ Fam102a^{flx/+}* mice, while periosteal perimeters were comparable to those of *Sp7-Cre⁺ Fam102a^{flx/+}* mice. We have included these new results in the revised manuscript (Supplementary Fig. 8; page 12, lines 173–176).

6. Figure 4: are there phenotypes in the global Fam102a null mice outside of bone?

Following the reviewer's suggestion, we examined the phenotypes of organs other than bone in global *Fam102a*-deficient mice. At 12 weeks of age, male *Fam102a*-deficient mice had similar body length and weight to control mice. Despite a significant decrease in pancreas weight and an increase in lung weight among the major organs, there was no apparent abnormalities in fasting blood glucose levels and histological analysis of the pancreas and lungs. Therefore, we conclude that these changes in organ weight did not affect bone metabolism. We have added these new results in the revised manuscript (Supplementary Fig. 9b,c; Supplementary Table 1; pages 13–14, lines 190–196).

7. Figure 5g: quantification should be performed for the Runx2 immunofluorescence data

Following the reviewer's suggestion, we quantified Runx2 expression in immunocytochemistry in Fig. 5f (formerly Fig. 5g). Our analysis revealed that the ratio of Runx2 expression in cytoplasm/nucleus was significantly increased in *Fam102a*-deficient osteoblasts, supporting our conclusions that nuclear Runx2 expression is decreased in *Fam102a*-deficient osteoblasts. We have included the new result in revised manuscript (Fig. 5f).

8. Figure 6: it is unclear how RNA-seq helps to identify direct 'clients' of Fam102a, and why it is necessary to study factors other than Runx2 to explain the phenotype observed here

We regret that this point was not made clear in the previous version. We showed that the significant decrease in bone mass in *Fam102a*-deficient mice was not correlated with the severity of the *Fam102a*-deficiency-induced decrease in nuclear Runx2 expression. This finding led us to investigate additional mechanisms by which loss of *Fam102a* reduces osteoblast differentiation other than Runx2 (see the details in our response to the Major point #2 by Reviewer #1). Since *Fam102a* seems to control the nuclear translocation of transcription factors by binding with *Kpna2*, we performed RNA-seq analysis to explore the pathways that are potentially regulated by *Fam102a*. When we focused on the transcription factors that were particularly affected by the loss of *Fam102a*, the most significantly downregulated gene was *Rbpjl*. Therefore, we did not employ RNA-seq to identify direct 'clients' of *Fam102a*. We apologize for any confusion this may have caused. Previous reports have shown that *Rbpjl* is capable of self-amplification in pancreatic acinar cells. Therefore, we speculated that *Fam102a* regulates the self-amplifying ability

of *Rbpjl* by affecting its nuclear translocation and tested the direct binding of Fam102a to *Rbpjl*. We have stated this point in the revised manuscript (page 18, lines 267–275).

9. Figure 8: as noted above, these results are interesting, but the relationship to Fam102a should be clarified

We examined the role of Fam102a in relation to *Rbpjl*-mediated osteoblast differentiation. Compared to Runx2 overexpression alone, both Fam102a and Runx2 overexpression led to enhanced *Sp7* promoter activity (Fig. 5c). We therefore tested the function of Fam102a on Runx2-induced activation of *Sp7* promoter in *Rbpjl*-deficient osteoblasts. Overexpression of Fam102a only slightly enhanced Runx2-mediated *Sp7* promoter activity in *Rbpjl*-deficient osteoblastic cells as compared to overexpression in wild-type cells. These results suggest that the effect of Fam102a overexpression is attenuated by the lack of *Rbpjl*, which functions downstream of Fam102a in osteoblasts. Therefore, these results support our conclusion that *Rbpjl* plays a crucial role in the regulation of osteoblast differentiation by Fam102a. We have included these new results in the revised manuscript (Supplementary Fig. 18d; page 23, lines 352–354).

10. Overall, the manuscript would benefit to some degree from editing for grammar/syntax

We regret any mistakes we may have made in our English writing. Since we are not native English speakers, we had our manuscript edited by an expert. Now we hope that our manuscript is clear enough for publication.

Reviewer #2:

The authors provide a comprehensive study regarding the function of Fam102a in bone remodeling cell types. They essentially confirm previous data regarding the role of this molecule in osteoclast differentiation, which explains, together with the low bone mass phenotype of mice ubiquitously lacking Fam102a, why the authors focused on the function of Fam102a in osteoblasts. They show that Fam102a is involved in the nuclear translocation of key transcription factors (Runx2, but also *Rbpjl*), which eventually enhances the expression of Osterix (encoded by the *Sp7* gene) to promote bone formation. Overall, this is a convincingly performed and presented study. However, the authors should be much more careful with their statements about the potential clinical relevance. In fact, they should focus less on

osteoporosis as a major health problem, but rather discuss what is known about functions of Fam102a (also known as Eeig1), not only in osteoclastogenesis (Refs 16,17), but also in other cell types.

We are very grateful to the reviewer #2 for the positive comments and constructive suggestions. Please find our responses to the comments raised below. We hope that the manuscript has been improved to the satisfaction of the reviewer.

1) One major goal of the authors was to identify a molecule required for differentiation of both, osteoclasts and osteoblasts. Since it is quite obvious, however, that these two cell types are entirely different in terms of progenitor cells, morphology, mode of action and regulatory molecules, the relevance of such a research question is at least debatable. Moreover, whereas the authors state twice that there are only few reports identifying factors required for the differentiation of both bone remodeling cell types, they don't include the respective references. It would be very informative to know, which examples the authors are referring to.

We apologize for the lack of detail in the introduction. As the reviewer noted, osteoclasts and osteoblasts have distinct differentiation mechanisms. Therefore, a careful approach is required when searching for shared regulatory mechanisms between these two cell types of different origin. However, as mentioned in the main text, certain factors and pathways, such as NFAT and Wnt signaling, regulate the differentiation of both cell types. Current treatments for osteoporosis fall into two categories; antiresorptives and anabolics, but both have limitations, including the risk of atypical fractures and complications associated with long-term use. Therefore, the next generation of osteoporosis treatment is expected to be an approach that targets both osteoclasts and osteoblasts. To develop such a treatment, it is critical to identify molecules involved in the differentiation of both osteoclasts and osteoblasts, as explored in this study. We believe that this research will serve as the basis for future drug discovery. In line with the reviewer's suggestion, we have added and revised the relevant section in the introduction (page 4, lines 52–60). We hope that the reviewer will find these revisions satisfactory.

2) Especially given the striking difference in TRAP staining shown in Supplementary Figure 1c, it would be advantageous to display the expression values for known markers of osteoclastogenesis and osteoblastogenesis as obtained by the Gene Chip analyses shown in Fig. 1a. Similarly, the authors should provide expression levels for known osteoblast markers as obtained by the RNA sequencing experiment shown in Fig. 5a.

As suggested, we have included the expression levels of known marker genes in the comprehensive gene expression analysis shown in Fig. 1a and 5a. Genes associated with osteoclast differentiation were downregulated in osteoclasts derived from *Ctsk*^{Cre/+} *Nfatc1*^{flox/flox} mice; similarly, genes associated with osteoblast differentiation were downregulated in osteoblasts lacking *Runx2* or *Fam102a*. We have included these new data in the revised manuscript (Supplementary Fig. 2a,b; Supplementary Fig. 11a; page 7, lines 95–97; page 8, lines 100–103; page 15, lines 212–213).

3) Since a mouse deficiency model for Fam102a/Eeig1 has been described previously, it would be important to better illustrate the targeting strategy used by the authors for the present study. It is also not specified, how exactly the global deletion was achieved.

We apologize for the lack of detail in the Methods section. In response to the reviewer's suggestion, we have illustrated the targeting strategy used to generate *Fam102a*-deficient mice and included it in the Supplementary Fig. 6 and the Methods section.

4) It would be informative if Fig. 8a could also display Runx2 binding sites in the Sp7

We appreciate the reviewer's careful consideration of our manuscript. Based on the reviewer's suggestion, we have depicted the Runx2 binding site in the *Sp7* promoter region in Fig. 8a.

5) The statement that “elucidation of the Fam102a function may lead to new options for the treatment of age-related osteoporosis” needs to be supported by additional information. How should such a treatment be pursued, if Fam102a is located intracellularly? Are there known agonists for Fam102a or proteins with similar function? In which tissues and cell types is Fam102a expressed? Are there any other phenotypes, for instance reduced pancreas weight, in mice ubiquitously lacking Fam102a? And finally, should a molecule promoting the differentiation of both, osteoclasts and osteoblasts, be considered as a drug target to counteract osteoporosis? In any case, as there is no human genetic evidence so far to support a key role of Fam102a in bone remodeling, the clinical relevance of mouse and cell culture data should not be over-estimated.

We apologize for overemphasizing the clinical relevance of our data in the Discussion section. We believe that Fam102a has the potential to become a novel therapeutic target

for normalizing bone metabolism in metabolic bone diseases. However, as the reviewer pointed out, we should not overestimate the clinical relevance of the mouse and cell culture studies due to the lack of sufficient scientific evidence in humans. To improve the accuracy of our manuscripts, we have removed the inaccurate descriptions in the Discussion section. Additionally, according to a public database, *Fam102a* is expressed in various tissues, including osteoclasts (Supplementary Fig. 3). Moreover, we examined the phenotypes of organs other than bone in global *Fam102a*-deficient mice. At 12 weeks of age, male *Fam102a*-deficient mice had similar body length and weight to control mice. Despite a significant decrease in pancreas weight and an increase in lung weight among the major organs, there was no apparent abnormalities in fasting blood glucose levels and histological analysis of the pancreas and lungs. Therefore, we conclude that these changes in organ weight did not affect bone metabolism. We have included these new results and revised the relevant sections (Supplementary Fig. 9b,c; Supplementary Table 1; pages 13–14, lines 190–196).

Responses to reviewers' comments on NCOMMS-24-10523A

Reviewer #1:

We are grateful to the reviewer for acknowledging our revisions. We greatly appreciate the additional suggestions. We hope that the manuscript has been sufficiently improved to the satisfaction of the editor and reviewers.

All previous comments have been addressed in a satisfactory manner. One minor point: the immunoblot in panel 6c appears to be cropped between lanes 3 and 4. The figure should be modified accordingly.

We apologize for any confusion this may have caused for the reviewer. In Figure 6c, the immunoprecipitated proteins from co-immunoprecipitation assay were loaded in lanes 1 to 3, and the input samples were loaded in lanes 4 to 6. The fact that different membranes were used in lanes 1 to 3 and 4 to 6 may have led the reviewer to think that the images were cropped. To avoid any misunderstandings, we have realigned the positions of the immunoblot images.